# Sea Level Rise in Europe: Adaptation measures and decision-making principles

**Giulia Galluccio[1], Jochen Hinkel[2,3], Elisa Fiorini Beckhauser[1,4], Alexander Bisaro[5], Rebeca Biancardi Aleu[1], Pierpaolo Campostrini[7], Maria Florencia Casas[1], Océane Espin[1], and Athanasios T. Vafeidis[6]**

[1]CMCC Foundation – Euro-Mediterranean Center on Climate Change, Milan, Italy
[2]Global Climate Forum (GCF), Berlin, Germany
[3]Resource Economics Group, Thaer-Institute of Agricultural and Horticultural Sciences,
Humboldt Universität zu Berlin, Berlin, Germany
[4]Department of Legal Sciences, University of Salento, Lecce, Italy
[5]Adapt 3E, Berlin, Germany
[6]Department of Geography, Christian-Albrechts University Kiel, Kiel, Germany
[7]CORILA, Venice, Italy

**Correspondence:** Giulia Galluccio (giulia.galluccio@cmcc.it) and Alexander Bisaro (bisaro@adapt3e.org)

**Abstract.** Sea level rise (SLR) will increasingly impact European countries in the coming decades, posing challenges for coastal decision-making and the design and implementation of adaptation measures to address coastal risks. The impact of SLR extends to its physical damages, encompassing socio-economic and environmental repercussions. European countries are engaged in the development and implementation of adaptation measures to bolster coastal resilience. While significant strides in SLR adaptation have been made in recent years, this paper aims to provide a catalogue of adaptation measures in European basins to guide their design and implementation and to present approaches suitable for supporting coastal adaptation decision-making and addressing uncertainty. The assessment of SLR adaptation measures in Europe is based on the cataloging of 17 measures following International Panel on Climate Change (IPCC) classification of *accommodate, protect, advance* and *retreat* responses to SLR, supplemented with sub-key types of measures, including socio-economic, physical and technological as well as nature- and ecosystem-based. Surveying the relevant literature on European sea basins, the paper shows that adaptation strategies on Europe's coasts constitute a mix of hard and soft measures, planning measures, policy developments and stakeholder and community engagements. Across all the basins, a common theme is the shift towards a combination of traditional engineering solutions with soft measures, including nature-based solutions, integrating local communities into decision-making processes and emphasising the importance of continuous monitoring and flexible management strategies. In addition, the context, decisions and experiences with coastal adaptation vary considerably across places and regions in terms of the time horizons considered, the scale of investments involved and the risk acceptance preferences of decision-makers and their constituencies. In this sense, the paper provides an overview of the common features of coastal adaptation decisions and the key aspects that need to be considered in coastal adaptation decision-making, i.e. considering multiple criteria and interests, implementing low-regret and flexible options, keeping future options open and factoring SLR into decisions that need to be made today.

# 1  Introduction

Global coastal systems are witnessing an increase in sea level rise (SLR), ocean acidification and rising ocean temperature, severely exposing people in low-lying areas to natural hazards and leading to significant environmental and socio-economic damages (Merkens et al., 2016). European coasts are subjected to an increase in sea levels and an increase in SLR adverse impacts, in particular coastal flooding, saltwater intrusion, coastal erosion and negative impacts on ecosystems and estuaries, affecting the ability of coasts to adapt to the changing climate (as demonstrated in Van De Wal et al., 2023).

A major concern for many countries is *how* to reduce exposure to SLR and enhance coastal resilience. For several centuries decision-makers have implemented traditional engineering solutions, herein referred to as grey options, as they dominated thinking and practice in coastal protection against SLR (Sancho, 2023; Kraus, 1996; van Koningsveld et al., 2008). A recent body of scientific evidence is proving that context-adjusted nature- and ecosystem-based solutions (i.e. green and blue options) as well as hybrid solutions can similarly reduce the risk of coastal flooding and erosion induced by SLR (Kuwae and Crooks, 2021).

Despite the growing attention placed on coastal adaptation, there is limited reporting of adaptation measures in the peer-reviewed literature and in policy documents, as they often present broad objectives rather than detail concrete measures. While systematic reviews have been done of global civil and environmental infrastructures of coastal adaptation to SLR (Nazarnia et al., 2020), of the role of protected areas in community adaptation in coastal areas (Ferro-Azcona et al., 2019), of studies performing socio-economic assessments of climate change adaptation in coastal areas (Riera-Spiegelhalder et al., 2023), of the limits of participation and co-production in climate adaptation within European coastal communities (Sartorius et al., 2024) and of public preferences regarding coastal adaptation measures (Mallette et al., 2021), European regional studies on adaptation solutions encompassing multiple types of measures – civil infrastructures, nature-based solutions or social, economic and institutional ones – are lacking. Besides, compliance with coastal laws by states and private actors is still overlooked in the scientific literature, despite being a critical aspect for addressing the impacts of sea level rise.

To facilitate climate action against SLR, the International Panel on Climate Change (IPCC) identifies four types of responses to SLR that guide countries in designing effective adaptation strategies: (i) *accommodate*, (ii) *protect*, (iii) *advance* and (iv) *retreat* (Oppenheimer et al., 2019). These represent four different approaches for adapting to natural hazards by reducing risks, exposure and vulnerability in low-lying coastal areas. Similarly, the European Environment Agency (EEA) developed the Key Type of Measures for Adaptation to Climate Change framework to report climate adaptation actions in EEA member countries. It has two categories of measures (*key types* and *sub-key types*), including socio-economic, physical and technological as well as nature- and ecosystem-based ones (Leitner et al., 2020). The advantage of using frameworks is that they help to standardise existing efforts in climate adaptation and capitalise on individual action for collective action while guiding the development of new efforts.

The contribution of this paper is twofold. First, in an effort to facilitate the diversification of local and national adaptation strategies portfolios for decision-makers, it collects and discusses 17 coastal adaptation measures implemented in European basins and provides a categorisation following the frameworks of the IPCC and EEA. Second, it presents approaches suitable for supporting coastal adaptation decision-making and addressing uncertainty. In doing so, it aims to fill the research gap within the coastal adaptation strategies landscape, to provide new analysis of and reflections on the existing adaptation measures in European basins, and to support decision-making.

As for the structure, Sect. 2 and its subsections present state-of-the-art SLR adaptation measures in Europe and aim to provide guidance for the design and implementation of adaptation policies in European basins. The section is further complemented by a series of in-depth analyses showcasing the implementation of adaptation measures in Venice, Italy, in Aveiro, Portugal, and in the Wadden Sea. Section 3 and its subsections first briefly review decision science terminology and then present key aspects that need to be considered in coastal adaptation decision-making, together with some example tools that can be used for addressing them.

# 2  Assessment of adaptation measures in Europe

A systematic scientific literature review was carried out, consisting of 247 scientific peer-reviewed articles, reports, policy documents and other grey literature to identify a list of adaptation measures, provide their description and find examples of best practices. The literature was collected through an iterative mixed-method approach (Fig. 1). First, 127 articles were identified using Web of Science Core Collection, searching the keywords "coastal adaptation" OR "coastal governance" AND "sea level rise" (topic) AND 2017–2023 (year published) AND Europe (topic). The review considered papers written between 2017 and 2023 to find the most up-to-date literature and provide emerging contexts and measures regarding SLR. Second, grey literature was included: 43 strategies, management and adaptation plans from different countries, regions and cities as well as 32 other sectoral reports and documents, comprising Maritime Spatial Planning country information. Third, 45 additional scientific studies were identified through references in peer-reviewed papers and included in the literature review. A selection of the literature was carried based on the following criteria:

the type of adaptation option (green, blue and grey), the sea basin (Mediterranean basin, North Sea, Black Sea, eastern Atlantic, Arctic basin and Baltic Sea) and type of impact (coastal flooding, saltwater intrusion, coastal erosion and negative impacts on ecosystems and estuaries). For a targeted collection of the literature, we have limited the search words. However, further research could be broadened to incorporate additional keywords such as "coastal strategy", "coastal defence", "adaptation to coastal flooding", "adaptation to coastal erosion", "adaptation to saltwater intrusion" and "adaptation of coastal ecosystems".

The main outcome of the literature review, which is represented in Table 1, is the collection and categorisation of 17 adaptation measures to SLR focusing on European sea basins and targeting four climate impacts: *coastal flooding, saltwater intrusion, coastal erosion and impacts on ecosystems and estuaries* (see Van De Wal et al., 2023). Table 1 lists the identified measures and provides information on the type of response, the sub-key type of measure (sub-KTM), the sea basin, the impact and the literature.

The top-level categorisation of adaptation measures is along the four main types of responses to SLR identified by the IPCC. First, accommodate measures involve preparing for and responding to coastal hazards. They include a range of responses, such as using early-warning systems, building flood-proof structures, managing groundwater and implementing insurance and policy instruments. Second, protect measures aim to reduce risks and impacts of coastal hazards through hard defence and soft defence measures. Additionally, nature- or ecosystem-based adaptation measures are also considered protect measures. Third, advance measures include strategies such as raising and advancing coastal land, e.g. by creating new raised ports, raising urban embankments and creating vegetated areas to promote natural land growth. Lastly, retreat measures include different adaptation measures, ranging from relocating human activities and infrastructure away from high-risk coastal areas to less vulnerable ones to restoring ecosystems by leaving coastal areas alone.

Adaptation measures are further categorised along the sub-KTM dimension developed by the EEA (Leitner et al., 2021). This categorisation is based on five main Key Types of Measures (KTM) and 11 sub-KTM, i.e. Governance and Institutional (policy instruments; management and planning; coordination cooperation and network) (see Bisaro et al., 2024), Economic and Finance (financing and incentive instruments; insurance and risk-sharing instruments), Physical and Technological (grey options; technological options), Nature Based Solutions and Ecosystem-based Approaches (green options; blue options) and Knowledge and Behavioural change (information and awareness raising; capacity building, empowering and lifestyle practices).

It should be noted that it can be difficult to draw clear distinctions when categorising measures, as the adaptation measures identified in the table can often be implemented at different levels of governance and at different spatial scales (see Bisaro et al., 2024). Moreover, some measures may in practice include activities across multiple sub-KTM and even combine multiple types of responses. For example, urban land raising (advance measure) may be appropriately combined with improved building codes (accommodate measure) in order to effectively reduce coastal risks, as in Hamburg's Hafen City (Bisaro et al., 2020). To ease the categorisation, the measures were classified based on the primary response and sub-KTM addressed.

The literature review shows that accommodate measures are the most widely discussed, followed by protect measures, advance measures and finally retreat measures. The most common sub-KTM is management and planning, followed by grey, green and blue options, insurance and risk-sharing instruments and technological options. The sea basins most covered in the literature are, respectively, the eastern Atlantic, the Mediterranean Sea, the North Sea and the Baltic Sea. Lastly, most measures focus on avoiding coastal flooding and erosion, while studies on ecosystems, estuaries and saltwater intrusion are very scarce. Based on the categorisation described above, the following section looks at each measure individually.

## 2.1 Types of responses to sea level rise

### 2.1.1 Accommodate

Accommodate measures include a range of biophysical, architectural and institutional responses. They do not directly prevent coastal impacts but rather mitigate coastal risks by reducing the vulnerability of coastal residents, ecosystems, human activities and the built environment, thus enhancing coastal communities' resilience. Accommodate is usually implemented in response to coastal hazards, coastal flooding, salinisation and other sea-borne hazards rather than directly to address SLR. The main advantage of accommodate measures is that they are generally both low-cost and highly cost-efficient in all contexts. This high cost–benefit ratio means that implementing them is much cheaper than not intervening (Oppenheimer et al., 2019). Accommodate measures can have additional advantages by producing and disseminating useful information, raising awareness of coastal risks among residents and promoting safer behaviour (Bongarts Lebbe et al., 2021).

Flood-proofing and raising buildings is an adaptation measure that involves the use of building techniques with specific designs and materials that are primarily aimed at flood risk reduction. Dry and wet-proof techniques have shown their effectiveness in reducing impacts of short periods of flooding (Ventimiglia et al., 2020). For long periods of high water, an appropriate measure is to raise buildings by elevating their height or constructing new ones at higher elevations (pile-dwelling construction or building on stilts). These can mitigate the risk of flooding and coastal inundation. Floating or

**Table 1.** Adaptation measures to sea level rise.

| Response | | Adaptation measure | Sub-KTM | Sea basin | Impact | References |
|---|---|---|---|---|---|---|
| Accommodate | 1 | Flood-proofing and raising buildings | Grey options | North Sea, Mediterranean Sea | Coastal flooding, coastal erosion | Dal Cin et al. (2021), Ventimiglia et al. (2020), Oppenheimer et al. (2019), Ministerio de Agricultura y Pesca, Alimentación y Medio Ambiente (2016) |
| | 2 | Adaptation measures to increase resilience of critical infrastructure | Grey options | Mediterranean Sea | Coastal flooding | Cavalié et al. (2023), Ward et al. (2020), Koks et al. (2023) |
| | 3 | Adaptation of ground-water management | Management and planning | North Sea | Coastal flooding, salt-water intrusion | 2023 Delta Programme (2023), MITECO (2020), Oppenheimer et al. (2019) |
| | 4 | Sustainable fisheries and aquaculture management | Management and planning | Baltic Sea | Impacts on ecosystems and estuaries | Payne et al. (2021), Oppenheimer et al. (2019) |
| | 5 | Climate risk insurance schemes | Insurance and risk-sharing instruments | Mediterranean Sea | Coastal flooding | Bednar-Friedl et al. (2022), Oppenheimer et al. (2019), Ministerio de Agricultura y Pesca, Alimentación y Medio Ambiente (2016) |
| | 6 | Consideration of climate change in credit risk and project finance assessments | Insurance and risk-sharing instruments | Mediterranean Sea | Coastal flooding | 2023 Delta Programme (2023), Oppenheimer et al. (2019), Netherlands Sovereign Green Bond (2023) |
| | 7 | Integration of climate change adaptation in coastal zone management plans | Policy instruments | Eastern Atlantic | Coastal flooding, coastal erosion | Bednar-Friedl et al. (2022), McEvoy et al. (2021), OECD (2019), Ministerio de Agricultural y Pesca, Alimentación y Medio Ambiente (2016) |
| | 8 | Early-warning systems and flood preparedness | Technological options | Eastern Atlantic, Mediterranean Sea | Coastal flooding | European MSP Platform (2022), Oppenheimer et al. (2019), Republic of Estonia (2017), Ministerio de Agricultura y Pesca, Alimentación y Medio Ambiente (2016) |
| | 9 | Develop a risk culture within the population | Information and awareness raising | Baltic Sea, eastern Atlantic, Mediterranean Sea | Coastal flooding | Zeng et al. (2020), Steljes et al. (2018) |

| Response | | Adaptation measure | Sub-KTM | Sea basin | Impact | References |
|---|---|---|---|---|---|---|
| Protect | 10 | Hard defence for coastal management (dams, dikes, levees, etc.) | Grey options | Eastern Atlantic, North Sea | Coastal flooding, coastal erosion | 2023 Delta Programme (2023), Del-Rosal-Salido et al. (2021), Egberts and Riesto (2021), Ministerio de Agricultura y Pesca, Alimentación y Medio Ambiente (2016), van Koningsveld et al. (2008), Hinkel et al. (2014), Lincke and Hinkel (2018), Tiggeloven et al. (2020), Vousdoukas et al. (2020), Hinkel and Nicholls (2020) |
| | 11 | Restoration and management of coastal ecosystems | Green and blue options | Eastern Atlantic | Impacts on ecosystems and estuaries, coastal flooding, coastal erosion | Moraes et al. (2022), Presidência do Conselho de Ministros (2019), Ministerio de Agricultura y Pesca, Alimentación y Medio Ambiente (2016), Buisson et al. (2012), Barbier et al. (2011) |
| | 12 | Beach and shoreface nourishment | Green and grey options | Eastern Atlantic, North Sea, Mediterranean Sea, | Coastal flooding, coastal erosion, impacts on ecosystems and estuaries | Tiede et al. (2023), 2023 Delta Programme (2023), Saengsupavanich et al. (2023), Sancho (2023), Mendes et al. (2021), de Schipper et al. (2021), Staudt et al. (2021), Pinto et al. (2020), Buisson et al. (2012) |
| | 13 | Other soft defence measures for coastal management (reloading littoral strips, cliff reshaping, polymer grids) | Green, blue and grey options | Eastern Atlantic | Coastal erosion | Oppenheimer et al. (2019), Presidência do Conselho de Ministros (2019), Buisson et al. (2012) |
| Advance | 14 | Raising and advancing coastal land | Green options | North Sea, eastern Atlantic | Coastal flooding, coastal erosion, impacts on ecosystems and estuaries | Van Den Hoven et al. (2022), Moraes et al. (2022), Laporte-Fauret et al. (2021), Bisaro (2019), Schuerch et al. (2018), Ministerio de Agricultura y Pesca, Alimentación y Medio Ambiente (2016) |
| Retreat | 15 | Planned relocation | Management and planning | Eastern Atlantic, Mediterranean Sea | Coastal flooding, coastal erosion, impacts on ecosystem and estuaries | Sayers et al. (2022), Government of Portugal (2021), OECD (2019), Thorsen et al. (2021), Schuerch et al. (2018), Van Den Hoven et al. (2022) |
| | 16 | Restricting new developments in flood-prone areas | Management and planning | North Sea | All | 2023 Delta Programme (2023), Oppenheimer et al. (2019) |
| | 17 | Managed realignment | Green and blue option | Mediterranean Sea Baltic Sea | Coastal flooding | Schuerch et al. (2018), Van Den Hoven et al. (2022), Thorsen et al. (2021), Bisaro et al. (2024) |

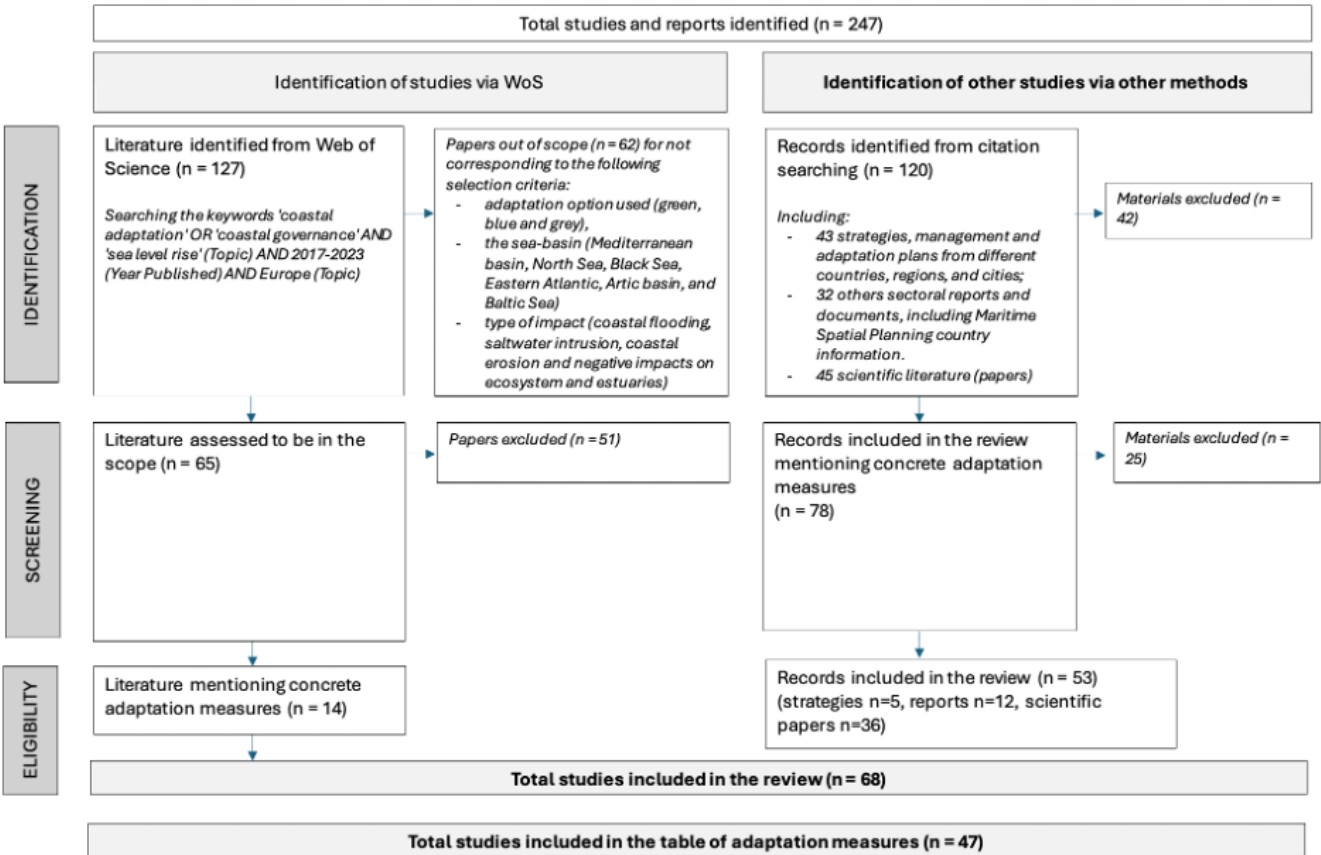

**Figure 1.** Methodological systematic review process.

amphibious buildings also offer the opportunity to float when flooding occurs for several months (Dal Cin et al., 2021). In the Netherlands, the latter technique has been tried with houses capable of adapting to different water levels (Oppenheimer et al., 2019). In Spain, the National Adaptation Plan focuses on the importance of using flood-proofed materials and building designs for critical infrastructure in coastal cities (Ministerio de Agricultura y Pesca, Alimentación y Medio Ambiente, 2016).

Increasing resilience of critical infrastructure involves solutions mainly composed of grey measures. Critical infrastructure is an asset that is essential for the maintenance of vital societal functions, mainly in the transport and energy sectors, e.g. ports, airports, highways or nuclear power plants. Critical infrastructure is often located near the coast, e.g. Schiphol Airport at 4 m below sea level in the Netherlands or Nice Côte d'Azur Airport in France at 3 m above sea level (Cavalié et al., 2023). The risks not only relate to the possible asset damages, but also concern the potential blockages and the disruption of economic activities that may result from infrastructure failure, as it could substantially increase the severity of the impact (Koks et al., 2023). This measure does not consist of precise preventive actions but instead involves methods to mitigate the risk of upholding the func-

tionality of the infrastructure. An example of how port authorities are dealing with climate change risks is provided by the government-led Ports of Spain, which manages 28 ports in the country. The Port Authority has adopted several measures to adapt to flooding and storm surges, including advanced early-warning systems, a new Spanish Ports Strategic Plan and the implementation of a Port Climate Change Observatory (see the box on "Climate change impacts and adaptation: status and challenges for the Spanish Ports system" in Bisaro et al., 2024). This critical infrastructure perspective is rarely addressed in the scientific literature and is more studied in the US than in Europe (Koks et al., 2023).

The sub-KTM management and planning include among others adaptation of groundwater management. Groundwater is an overexploited resource that is being used globally at an alarming and unsustainable rate, affecting its capacity to act as a natural buffer against coastal flooding (Ward et al., 2020). In turn, the conservation of groundwater reservoirs, the limit of water use and the optimisation of water reuse can avoid salinisation and increase the adaptive capacity of coastal areas. This calls for human activities conducive to the preservation and sustainable management of groundwater resources, in particular through improved land management practices in upper basins or in urban areas through

rainwater harvesting and the use of pervious pavements (Oppenheimer et al., 2019). For instance, the Freshwater Delta Programme in the Netherlands aims to prevent water shortage in the present and near future (2050) and includes comprehensive measures to maintain a healthy groundwater system, using spatial planning and other context-specific strategies (2023 Delta Programme, 2023). The multiple benefits of sustainable groundwater management make it both an accommodate measure and a protect measure. For a more extensive discussion of prevention and adaptation measures to limit groundwater salinisation, see Van De Wal et al. (2023).

The sub-KTM management and planning also include sustainable fisheries and aquaculture management. In recent years, the literature and political action in Europe have focused more on overexploitation of living marine resources than on climate change impacts, which is a severe issue, particularly in southern Baltic states (Payne et al., 2021). In studies that focus on climate-related drivers of fisheries and aquaculture, ocean warming and acidification are considered more influential than SLR (Oppenheimer et al., 2019). However, future projections of SLR and their implications for fisheries and aquaculture are an understudied area.

Climate risk insurance schemes can play an important role in enhancing coastal resilience and reducing vulnerability. These mechanisms can provide financial security to coastal communities and businesses to mitigate the financial impacts of loss events such as coastal flooding and storm events (see Bisaro et al., 2024). They have mainly been used in the context of agriculture and urban areas (Oppenheimer et al., 2019). The European insurance industry has developed flood-specific products, notably through risk-based flood insurance schemes that can induce risk-averse behaviour, and it is also investing in the field of risk analysis (Bednar-Friedl et al., 2022). Spain has developed specific insurance and reinsurance schemes like the "extraordinary risk insurance" for risks specifically deriving from SLR in coastal areas, including extraordinary floods and atypical cyclonic storms (Ministerio de Agricultura y Pesca, Alimentación y Medio Ambiente, 2016). More recently, governments have been funding post-disaster mechanisms, making flood insurance compulsory or taking on the role of reinsurer in public–private partnerships. Well-designed insurance schemes may also include measures such as reduced prices of the insurance if homeowners implement preventive adaptation measures, e.g. not keeping high-value items on the ground floor, which increase the overall effectiveness of insurance (Bednar-Friedl et al., 2022). However, when poorly designed, insurance schemes can also perpetuate the risk and incentivise maladaptation. An example is the provision of insurance pay-outs to rebuild assets in a location that is increasingly experiencing flood risk without proportionally increasing premiums. Moreover, increasing climate risks could put a strain on public budgets, leading to the withdrawal of support for publicly funding insurance and potentially reducing the availability or affordability of insurance products for poor households and some households in high-risk areas. Similarly, increasing risks may lead to decreased offerings of private insurances due to either insolvency or them exiting markets (Bednar-Friedl et al., 2022).

Addressing climate change in credit risk and project finance assessments is an accommodate measure as it orients investors towards projects that enhance adaptation. Consideration of climate change in credit and finance assessments can thus mobilise financing of specific projects against SLR through the public and private sectors, international climate funds and other innovative financing solutions. In 2019, the Netherlands issued the first certified Sovereign Green Bond by a European country (Netherlands Sovereign Green Bond, 2023). A large proportion of the bond proceeds was used to fund the Delta Programme, a sophisticated flood risk management system that enhances resilience to SLR and improves freshwater supply, among other benefits. The Delta Programme also has a specific Delta Fund, which is a separate item of the central government budget and includes EUR 21 billion available for the period 2023–2036 (2023 Delta Programme, 2023). An example of a tool for financing adaptation projects is to raise funds from the sale of newly generated lands coming from the implementation of advance measures (Oppenheimer et al., 2019). Another example is provided by the PIMA Adapta Plan for the Promotion of the Environment for the Adaptation to Climate Change in Spain, an operational tool that finances adaptation projects using emission rights, among others (MITECO, 2020).

The literature emphasises the key role of integrating SLR information into coastal adaptation strategies and plans. An illustrative case is Spain. Since 2004, Spain has prioritised climate change adaptation measures that protect its vulnerable coastline. The first National Plan for Adaptation to Climate Change (PNACC), approved in 2006, identified coastal impact assessment as a priority. The second (2009–2014) and third (2014–2020) PNACCs identified coastal zones and the development of a strategy for the adaptation of the coasts to climate change as a priority line of action, which was de facto adopted in 2016 (Ministerio de Agricultura y Pesca, Alimentación y Medio Ambiente, 2016). The current PNACC (2021–2030) foresees the development of risk analysis tools and the definition of adaptation initiatives on the coasts and at sea, the facilitation of coastal and marine adaptation through regulatory frameworks, the integration of coastal risks into plans and programmes as well as the fostering of institutional coordination and social participation for adaptation on the coasts and at sea.

SLR entered into innovative governance instruments that have been developed to overcome administrative barriers in coastal governance, e.g. the 2023–2027 Toulon Bay Contract which involves 40 local stakeholders in a decentralised, participatory and bottom–up approach to adapt to flooding and erosion risks (Métropole Toulon Provence Méditerranée, 2023). Further information on coastal governance instru-

ments is provided in the section "Equity and Social Vulnerability" in Bisaro et al. (2024).

The literature also stresses the importance of studying multiple time horizons and different scenarios of SLR. The effectiveness of some adaptation strategies has been compromised by the use of only a few scenarios and the use of a single time horizon as opposed to multiple ones (OECD, 2019). For example, in Venice's adaptation pathways, only shared socio-economic pathways SSP1–2.6 and SSP5–8.5 were considered without using intermediate scenarios. As such, once critical relative sea level thresholds are reached, the remaining upper limit will represent a low-likelihood but high-impact storyline (Bednar-Friedl et al., 2022). Similarly, if planning only accounts for the short term, they may no longer be adequate once the adaptation measures are finally completed, especially given that major permeant interventions may take a long time to implement (Bednar-Friedl et al., 2022).

The implementation responses to SLR have been facilitated by the advancement of predictive tools and cartographic techniques designed to forecast the extent and repercussions of such rise and the subsequent floodings (Mcleod et al., 2010). Technological options include early-warning systems and flood preparedness, and they support all types of responses to varying degrees. They are conventionally considered an accommodate measure because they allow people to remain in the hazard-prone area but help improve preparedness and response by providing advance warning in the face of imminent danger. However, early-warning systems are also used in other types of responses, such as in protection (in the case of mobile protection defences like the Thames Barrier and the MOSE barrier in Venice; see Box 1) and retreat (in the case of extreme events evacuating people) responses. They have short implementation times and low impacts on the environment, but their implementation and effectiveness largely depend on good forecasting, predictable hazardous events and definition of adequate early-warning indicators (Oppenheimer et al., 2019). Thus, they are less well suited to accommodating slow onset change. Spain's adaptation plan has examples of early-warning systems and also evacuation protocols, which are carried out in coordination with societal organisations as well as local communities affected by the dangers (Ministerio de Agricultura y Pesca, Alimentación y Medio Ambiente, 2016). Estonia offers another interesting case of actions aimed at improving knowledge of SLR and flood preparedness. Its strategy incorporates an accommodate measure to develop sea level forecasting systems for areas prone to coastal flooding (Republic of Estonia, 2017). As a result, Estonia has implemented a Maritime Spatial Plan for 2022, which includes a study of the expected SLR along the −3 m contour from the coast, specifically in the Pärnu Bay area (European MSP Platform, 2022).

Developing a risk culture within the population subcategorised as information and awareness raising relies on an understanding of how people perceive risk and act in particular ways (Zeng et al., 2020). This can be an effective adaptation measure as some of the basic requirements for successful collaboration in communities to manage and cope with extreme events are "culture of risk memory", "trust in scientific information and community" as well as trust in coastal authorities (Stelljes et al., 2018). This measure could equally be considered part of a long-term retreat measure because developing a risk culture prepares the population for potential future relocation.

### 2.1.2 Protect

Protect measures aim to reduce the risks and impacts of coastal hazards. These measures typically entail the construction and upgrade of hard and soft defences (OECD, 2019) but can also refer to restoration and management of coastal ecosystems.

Hard defence for coastal management includes the implementation and upgrade of physical structures such as dams, dikes, levees, groynes, breakwaters, artificial reefs, sea walls, jetties, storm surge gates, flood barriers and other types of defences. These are classified as grey measures that aim to prevent coastal erosion and flooding.

Hard defences have been very widely applied for centuries to prevent coastal erosion and flooding. The North Sea coastline of Belgium, the Netherlands and Germany is protected by dike systems complemented by other measures such as sand nourishment, dunes and surge barriers. Hard defences have also been implemented to counter relative SLR caused by land subsidence, such as areas with young sediments like the Italian Po Delta, the Netherlands and northern Germany (van Koningsveld et al., 2008).

Some advantages of hard defences are that they have long life spans, and their costs are reasonably well known and can be estimated. Generally, hard defences are highly effective at protection but generally leave a low risk of failure unless defences are built so wide that they cannot breach (De Bruijn et al., 2013). There are also economic motivations linked to the cost–benefit ratio of investments. Generally, hard protect measures are economically beneficial in urban areas as they have high cost–benefit ratios, and this has also been widely found to be true for 21st century SLR (Hinkel et al., 2014; Lincke and Hinkel, 2018; Tiggeloven et al., 2020; Vousdoukas, et al., 2020). For rural and less densely populated areas, hard protection is generally not economically beneficial, which suggests that alternative measures, in particular ecosystem-based measures or retreat, are often better solutions (Hinkel and Nicholls, 2020).

Negative consequences of coastal protection infrastructure include the need for ongoing maintenance and alterations in natural coastal dynamics, due to e.g. loss of plants and mosses, and hard defence measures can also negatively impact cultural heritage by changing the existing landscape (Egberts and Riesto, 2021). Some examples of this can be seen in the national adaptation plans of Spain (Ministerio

de Agricultura y Pesca, Alimentación y Medio Ambiente, 2016) and the Netherlands (2023 Delta Programme, 2023). An example of hard defence in the context of cultural heritage and landscape protection is the renowned MOSE system in Venice that after several decades of discussion and development entered into operation on 3 October 2020 (see Box 1 below).

Box 1: The MOSE system for protecting Venice and its lagoon

On 4 November 1966, due to an extreme and unexpected meteorological event, the water level reached 194 cm above the historical mean sea level and remained above 110 cm for 22 h. On 16 April 1973, the Italian Parliament promulgated the first Special Law for Venice, declaring the protection of Venice and its lagoon to be of primary national interest. Figure 1 demonstrates how the frequency of floods in the city increased from 30 to 95 events per decade, 1970–1079 and 2010–2019 (Fig. 2).

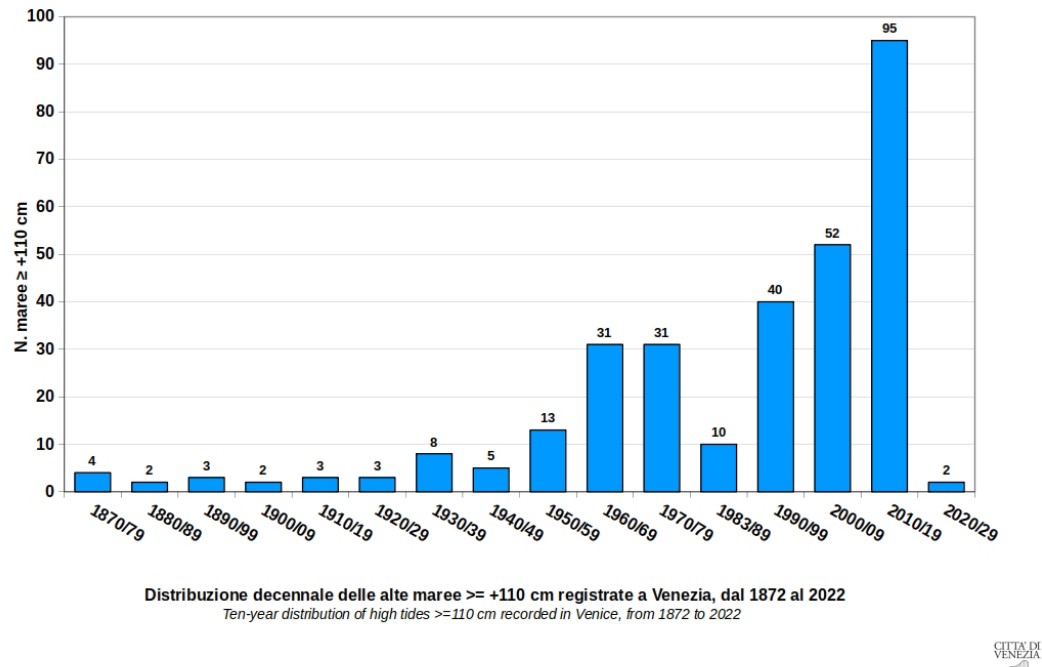

**Distribuzione decennale delle alte maree >= +110 cm registrate a Venezia, dal 1872 al 2022**
*Ten-year distribution of high tides >=110 cm recorded in Venice, from 1872 to 2022*

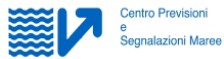 Centro Previsioni e Segnalazioni Maree

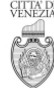

**Figure 2.** Number of city flooding events in Venice per decade. The distribution indicates the number of events with a sea level higher than 110 cm. The original source of this figure is the Municipality of Venice – Centro Previsioni e Segnalazioni maree.

After a long period of discussions, prototype testing and design revisions, the construction of the MOSE barriers began in 2003 and became operational for the first time on 3 October 2020, effectively protecting the centre of Venice and all the lagoon settlements. The MOSE barriers are an essential part of a much wider safeguarding approach that includes littoral island defence, adaptation measures in the urban settlements, ecological and morphological restoration of the lagoon (the largest in the Mediterranean Sea, ca. 550 km$^2$), de-pollution and defence measures in the lagoon basin (2068 km$^2$).

The "Venice SLR defence approach" is a mixture of protect and accommodate interventions which represent a continuation of what the Serenissima Republic of Venice did in its millenary history. The narrow littoral islands of Pellestrina and Lido, which separate the Venice Lagoon from the Adriatic Sea, were made of sandbanks when the lagoon was formed around 6000 years ago. However, already 7 centuries ago, the need to protect the coastal settlements from sea storms led the Republic of Venice to develop a complex defence system made of wooden poles ("palade") that were regularly renovated. In the 18th century, this defence was replaced by massive stone sea walls ("murazzi") placed on the shore. Since 2000, the ancient sea walls have been repaired and reinforced by a new shore in the form of gyrons built in front of them, with sand taken from the Adriatic Sea. This is the largest confined sand nourishment that occurs in Europe (Figs. 3 and 4).

The MOSE steel barriers placed at the lagoon's inlets can provide a complete closure of the lagoon from the sea, for a total length of 1.56 km divided into four arrays. They can guarantee a difference of 2 m between the lagoon and the sea level offshore, maintaining the level of the lagoon at the safe level of 100 cm above sea level during storm events of up to 300 cm (the maximum event ever measured is 204 cm). Each of the 78 floodgates is 20 m wide and varies its length according to the depth of the four inlets.

They normally lie inside big concrete caissons placed on the seabed, connected by two hinges on one end and filled with water. To close the barrier, the air is pumped into the gates by compressors, allowing them to float at the desired angle for

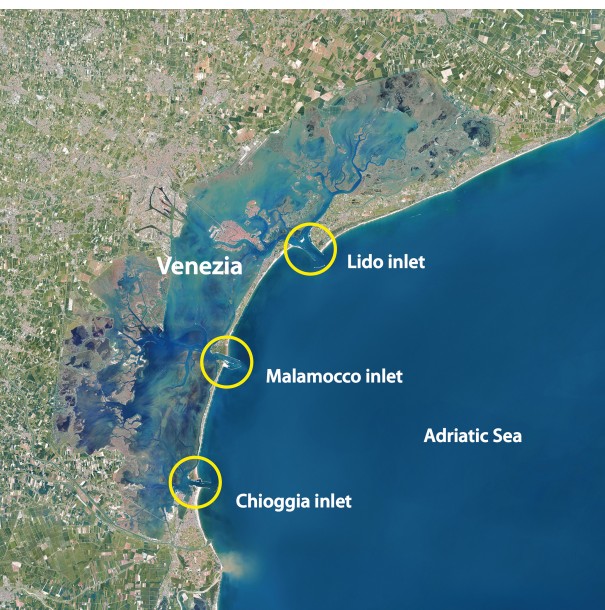

**Figure 3.** The Venice Lagoon and its three inlets. The original source of this figure is the Consorzio Venezia Nuova – Concessionaire of the Ministry of Infrastructure of Italy).

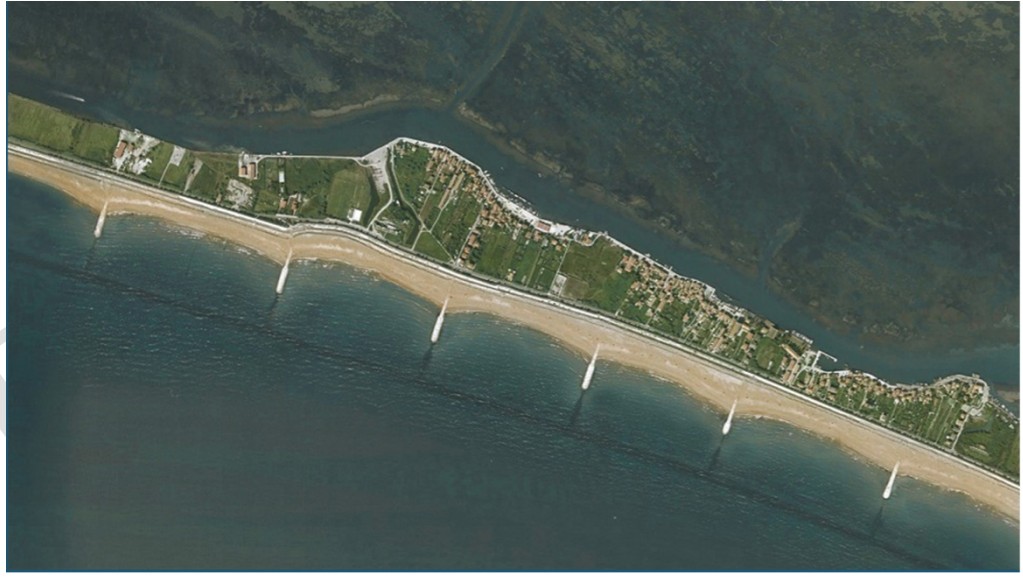

**Figure 4.** The new shore realised in front of the old murazzi on the island of Pellestrina. The original source of this figure is the Consorzio Venezia Nuova – Concessionaire of the Ministry of Infrastructure of Italy.

closure. Each gate floats independently of the others to avoid the risk of stress concentration that a single, longer barrier might experience (Fig. 5).

After some tests (Fig. 6), the MOSE barriers became operational for the first time on 3 October 2020 and in the first three winters operated 50 times, effectively protecting Venice from floodings, including severe ones (Fig. 7).

The closure of the lagoon should be kept to a minimum, for both ecological and economic reasons. The protection strategy foresees the raising of the city's pedestrian walkways to a minimum level of 110 cm above sea level. In fact, throughout its history, Venice has constantly raised the level of its buildings to cope with the relative SLR (eustacy and subsidence). In the last century, cultural heritage and landscape protection together with a faster SLR made these adaptation measures harder to

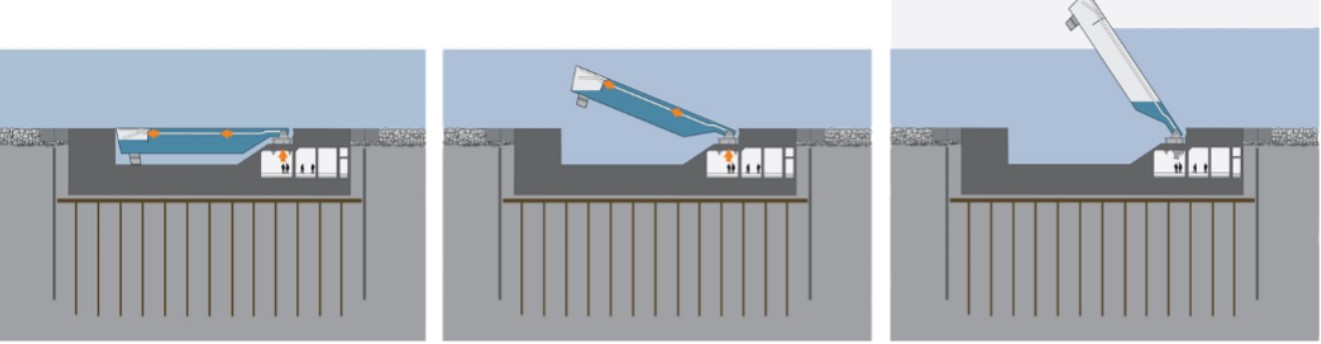

**Figure 5.** MOSE barrier functioning scheme. The original source of this figure is the Consorzio Venezia Nuova – Concessionaire of the Ministry of Infrastructure of Italy.

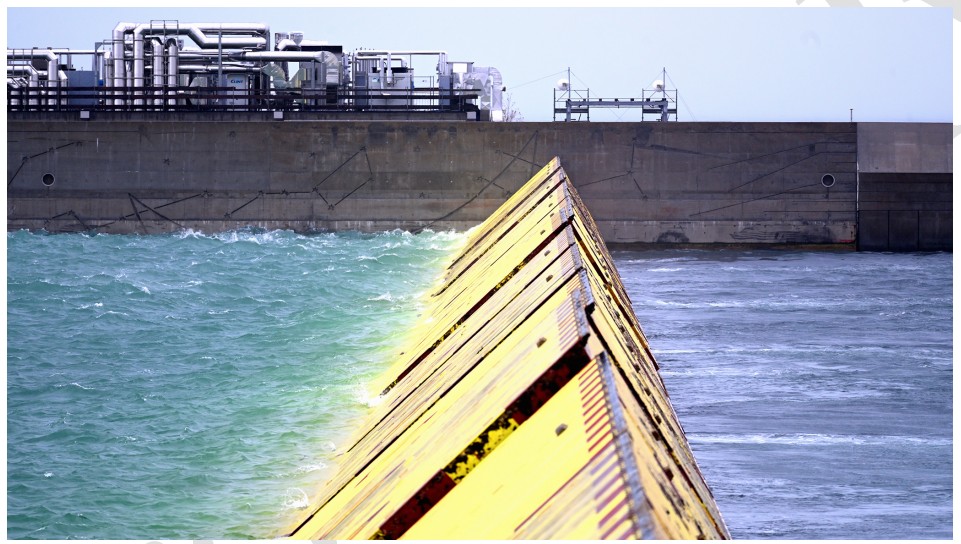

**Figure 6.** MOSE barriers on the Lido during a storm on 15 November 2020. In the picture the sea is on the left and the lagoon is on the right.

(a)                                                    (b)

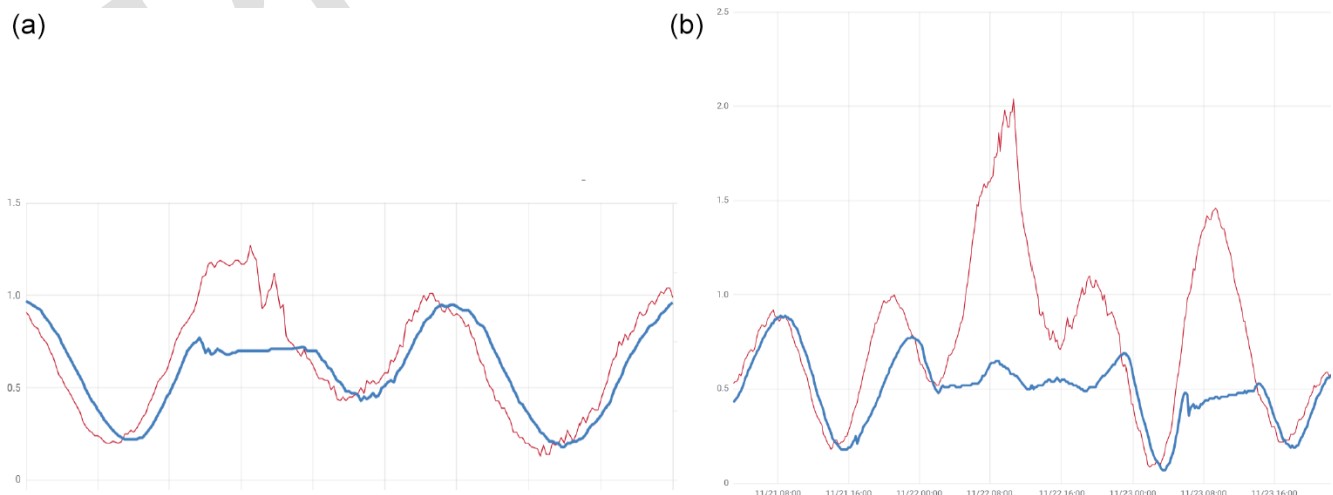

**Figure 7.** Sea level in the Adriatic (red) and inside the lagoon (blue) during the events of 3 October 2020 **(a)** and 22 November 2022 **(b)**.

implement. However, since the early 2000s Venice has continued to raise the level of the public pavements. Piazza San Marco represents a special case because of the presence of relevant artefacts placed at a much lower altimetric level.

In this case, an "impermeabilisation" strategy has been chosen, which consists in raising the level of the entire island of San Marco to 110 cm and in revising all the rainwater drains by installing suitable valves. These complex works are underway and will take several years to complete; in the meantime, in order to protect the most important monument, St. Mark's Basilica, from further saltwater intrusion, a glass barrier has been erected in front of the basilica facing the piazza (Fig. 8).

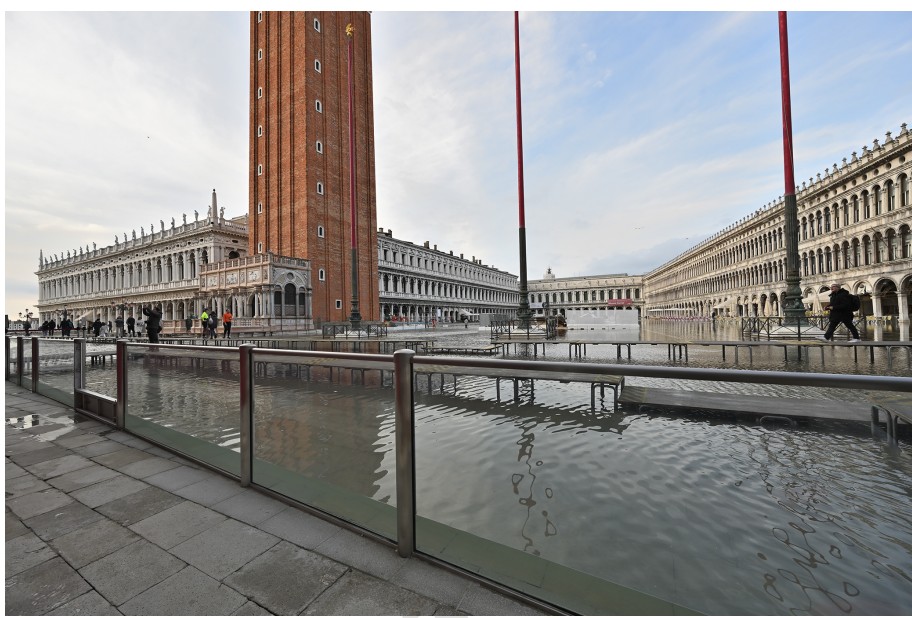

**Figure 8.** The glass barriers in front of St. Mark's Basilica provide effective protection, also from minor "acqua alta" events.

Once the main problem has been given a solution, other issues will continue to challenge science and policy.

As the rise in sea level continues, the frequency of barrier closures will increase: managing a regulated lagoon requires specific observational and modelling tools to be kept up to date. Further de-pollution and morphological interventions against salt marsh erosion are also needed.

It is well known that the paradigm of mobile barriers works up to a 50–60 cm SLR; above this threshold, these gates will be permanently closed and a different protection scheme should be provided. What this new system will be has not been discussed yet. In the coming decades, however, Venice will continue to be a multi-disciplinary and transdisciplinary laboratory for testing SLR adaptation measures for the whole world.

Soft defences for coastal management include different types of green, blue and grey options. One major difference between hard and soft protect measures is their respective impacts on natural sedimentary dynamics and equipment reversibility (Buisson et al., 2012). Two main examples of soft defences are dominating the discourse and are being extensively used in practice. First, the restoration and management of coastal ecosystems are common green and blue options used as an alternative to traditional approaches. Coastal vegetated ecosystems and biogenic reefs can self-adapt to SLR through different mechanisms (Moraes et al., 2022). These types of measures help to reduce erosion and flooding, in addition to providing a habitat for numerous species and other environmental benefits for local ecosystems (Barbier et al., 2011). Examples can be found in Spain (Ministerio de Agricultura y Pesca, Alimentación y Medio Ambiente, 2016), Portugal (Presidência do Conselho de Ministros, 2019) and France (Buisson et al., 2012). This latter study shows how France successfully restored marshes and other vegetated ecosystems, protecting against wave energy and limited erosion and sediment accumulation. In the UK, the creation, restoration and enhancement of estuarine, coastal and marine habitats are funded through the Environmental Land Management (ELM) scheme. One initiative in this scheme is Restoring Meadows, Marshes, and Reefs, which aims to restore at least 15 % of three priority habitats by 2043, providing support to farms to restore habitats along the coasts and support upstream improvements (Department for Environment, Food and Rural Affairs, 2023).

Second, beach and shoreface nourishment is the artificial supply of sand and occasionally gravel or small pebbles to increase coastal sediments. This expands the sand volume or width of the beach, allowing it to counter coastal erosion and sometimes to advance seawards (de Schipper et al., 2021). Providing beach space is beneficial for tourism and recreational activities (Mendes et al., 2021). The objective of this nourishment is to compensate for the littoral imbalance caused by natural erosion and anthropogenic impacts (Buisson et al., 2012). In the literature, the difference between beach nourishment and shoreface nourishment is mainly related to the location of sand placement, which may be, respectively, on the subaerial beach (above-water beach) or the subtidal beach (submerged near-shore beach profile) in the form of an underwater mound (Mendes et al., 2021). The materials are dredged from offshore and inland sources, including nearby navigation channels. For example, the Lisbon Port Authority regularly maintains the outer Tagus estuary navigation channel by dredging sand that can be used for beach nourishment (Sancho, 2023).

Beach nourishment has been applied more extensively in Europe since the 1990s. In particular, in the eastern Atlantic Ocean the increase in the number of beach nourishments has been accompanied by a reduction in the number of hard coastal structures, contributing to improvements in coastal sediment management (Pinto et al., 2020). In Portugal, an extensive beach nourishment programme was carried out in the framework of a coastal management master plan between 2007 and 2019 (Mendes et al., 2021; Pinto et al., 2020). The programme placed $4.5 \times 10^6$ m$^3$ of sand along a 3.8 km northern shoreline (Sancho, 2023). In Spain, the Adaptation Plan envisions the regeneration of beaches and artificial dune systems to reduce erosion and revitalise coastal ecosystems. As part of the Adaptation Plan, in the sandy area of Liencres, several interventions have been made to restore one of the largest dune systems of the Cantabrian Sea (Ministerio de Agricultura y Pesca, Alimentación y Medio Ambiente, 2016). In the Netherlands, Tiede et al. (2023) studied the changes in shoreline and coastal developments using satellite data of a sand nourishment initiative. The study compares images from the natural evolution period (1984–1990) and the recent nourished period (1996–2022), where approximately half of the sandy transects were nourished regularly in combination with small groynes to support the project (see Van De Wal et al., 2023). In brief, the study showed an increase in the share of stable or accreting transects from 67 % to 89 %, while the share of eroding segments fell by 20 % (Tiede et al., 2023). Similarly, the Wadden Delta Programme includes different operations of sand nourishment on the North Sea side of the Wadden Islands, protecting them against flooding and also preserving ecosystem functions (2023 Delta Programme, 2023).

Nourishment is a flexible and fast coastal management option that is adaptable to changing conditions, remaining relatively cheap even if nourishments have to be repeated. However, the recent literature questions the sustainability of sand nourishments (Saengsupavanich et al., 2023; Staudt et al., 2021). Criticisms stress the environmental impacts in both sediment extraction and at nourishment sites, in particular in relation to the destruction of habitats, disruption of bird and other animal nesting, coverage and subsequent suffocation of benthic organisms, the increase in water turbidity and shifts in median grain size and grain-size distribution depending on the chosen material. In addition, large uncertainties in the long-term ecological and geomorphological impacts of nourishment remain (Staudt et al., 2021).

Other examples of soft defence measures include the use of geotextile structures as sand containers, the creation of artificial reefs to reduce wave energy and prevent beach erosion, as well as plant debris cover, windbreaks and plantations (Buisson et al., 2012). For instance, hydraulic pilings made of wooden rods vertically planted in the sediment at regular intervals limit sedimentary transport and favour beach stability in pilot studies in France (Buisson et al., 2012). Another example of a soft measure is cliff strengthening and stabilisation, which includes green and grey options that focus on reducing erosion and enhancing natural protection along coastal cliffs. This includes a range of techniques such as reloading littoral strips to compensate for sediment imbalances caused by marine erosion, cliff reshaping, drainage systems and the use of anchoring elements like

bolts, tie rods, polymer grids, pinned nets and rip-rap strips (Buisson et al., 2012). This category of measure is employed in several countries, such as Croatia (Omiš) (Oppenheimer et al., 2019), Italy (Marche) (Addressing coastal erosion in Marche region, Italy) and Portugal (Presidência do Conselho de Ministros, 2019).

### 2.1.3 Advance

Raising and advancing coastal land (2024) has a long history of use to protect communities from natural hazards. Only recently has advance become a response to SLR on its own (Pörtner et al., 2019). Advance measures for coastal management include all those solutions that create or advance new land by expanding into the sea or ocean. Advance measures may be green or grey and mainly address coastal flooding, coastal erosion and biodiversity loss. Grey land reclamation emerges as an adaptation measure, particularly in high-value urban areas in Europe and globally (Bisaro et al., 2020). Raising and advancing coastal land (2024) is being pursued in major coastal cities, where new ports, harbour areas and safer urban embankments have been created in raised areas (Bisaro, 2019). At the global level, the most common land uses in reclaimed spaces are port extensions, exemplified by the two major ports in the Netherlands, Rotterdam and Amsterdam, which reclaimed 1106 and 337 ha, respectively, between 2000 and 2020 (Sengupta et al., 2023). Advance measures can also be ecosystem-based by including measures based on conservation and restoration of sediment systems, coral barriers or coastal vegetation by applying several techniques, such as excavation of foredune notches, dune thatching, dune grass planting, dune fencing or hybrid combinations of a dike core in a dune (Oppenheimer et al., 2019). For instance, in south-western France the excavation of foredune notches re-established an ecomorphological dynamic promoting landward sand transport and foredune landward translation, without threatening biodiversity.

### 2.1.4 Retreat

Retreat includes measures focused on reducing the level of exposure to coastal hazards by relocating human activities, infrastructure or even cities from highly exposed to less exposed areas. Retreat necessitates rethinking the entire coastal system as well as accepting that particular assets will need to be removed entirely (Bongarts Lebbe et al., 2021). The advantage of these types of measures is their effectiveness in both low- and high-risk coastal areas. However, they are solely applicable in regions with low population density (Oppenheimer et al., 2019). Retreat incorporates a wide range of measures mostly categorised as management and planning. Retreat measures have been implemented in various European sea basins, e.g. in the eastern Atlantic Ocean, the Baltic Sea and the Mediterranean Sea.

Planned relocation applies to individuals and critical assets, including the removal of existing hard infrastructure (OECD, 2019). This measure involves the governance and institutional planning behind the relocation of activities from high-risk areas, land acquisition and the expropriation of operations. Deciding to relocate a community has complex trade-offs: on the one hand there is an opportunity to reduce potential damages and meet the different needs and conditions of the community and, on the other hand, there are the high costs and direct impacts on people's lives, which require extensive engagement with the community and clear incentives (Sayers et al., 2022; OECD, 2019). For instance, approximately 30 % of England's coastline is likely to be under increasing pressure by the 2050s, affecting more than 120 000 properties, and a large but still unknown proportion of these properties will need to be relocated (Sayers et al., 2022). Another example is provided by Portugal, which has reported to the European Commission several measures that the country is implementing to manage the risk of SLR, including the progressive removal of constructions that are located in flood-critical territories along the coastline through spatial planning instruments (Government of Portugal, 2021).

Restricting new developments in flood-prone areas and defining setback zones is an approach to support planned relocation. An example is the Dutch Freshwater Delta Programme that spatially restricted development based on fluctuation levels (2023 Delta Programme, 2023). These flood-prone areas can be replaced with marshes or activities like aquaculture or salt-tolerant cultivation areas (Oppenheimer et al., 2019). The governance of flood-prone areas is also addressed in the Protocol on Integrated Coastal Zone Management (ICZM) of the Barcelona Convention (UNEP, 1995) – the main regional legally binding Multilateral Environmental Agreement in the Mediterranean, which entered into force in the European Union in 2011 after ratification. Article 8 of the protocol identifies a setback zone of a minimum of 100 m in width from the shoreline as a measure to protect coastal settlements and infrastructure from adverse impacts and is the first international legal instrument to require the use of coastal setback zones. Notably, the protocol links setback zones with adjacent areas such as wetlands and natural forests, which allows for the restoration of biodiversity and can serve as nature-based solutions (NbS) to adapt to the effects of climate change (Adriadapt, 2022).

An emerging option is managed realignment, a coastal adaptation strategy that entails the landward relocation of coastal defences to allow previously protected areas to restore tidal exchange and coastal habitats. A successful example of managed realignment in European basins, and the first large-scale example in Denmark, is the restored Gyldensteen Coastal Lagoon in the western Baltic Sea, where the ecological status improved and species richness increased after 5 years (Thorsen et al., 2021). Managed realignment as an

adaptation strategy for the Ravenna coastline in 2100 can be found in Box 2 in Bisaro et al. (2024).

## 2.2   Limits and trade-offs of adaptation measures

The adaptation measures discussed in the preceding section are generally subject to trade-offs that should be considered when planning coastal adaptation. While accommodate measures offer benefits such as cost-effectiveness and immediate relief, the financial cost of implementing these measures can be a challenge for some communities. Protect measures provide important risk reduction benefits. However, they can severely disrupt natural coastal processes and harm marine life. Even soft protection or advance measures can have similar, localised ecological effects (for example, altering sediment transport patterns may unintentionally lead to erosion in neighbouring regions). While sea walls provide coastal protection, they can also exacerbate erosion by affecting the entire ecosystem and thus diminishing the ability of the system to respond naturally to different conditions (Rijn, 2011). These measures may also impact cultural heritage sites and alter coastal areas in addition to requiring high maintenance costs. Lastly, retreat measures potentially displace entire communities and can involve the loss of assets and business activities (e.g. tourism-related activities). They therefore generally require complex governance and coordination among multiple stakeholders and are limited to regions with low population density. To accurately analyse existing trade-offs, understanding the effectiveness and feasibility of these measures is important. Currently, there is a critical literature gap in this regard. Information is lacking on the effectiveness of measures in reducing risk and the economic, technological, institutional, socio-cultural, geophysical and ecological feasibility of implementing them. Existing analyses of effectiveness and feasibility are typically undertaken for particular types of responses at the global level rather than for individual measures. There is thus a scientific need to evaluate the effectiveness and feasibility of individual measures and in context-specific cases. This represents a research gap that, if addressed, could advance knowledge and significantly contribute to the field of coastal adaptation.

Finally, while the identified measures can help communities and governments to adapt to the challenges posed by SLR, addressing SLR in coastal areas requires careful consideration of the trade-offs associated with accommodate, protect, advance and retreat measures. In an effort to minimise the trade-offs and to provide a multi-faceted, integrated and sustainable solution to rising sea levels, novel approaches combine more than one adaptation measure and develop hybrid solutions (see Box 2).

Box 2: The role of hybrid solutions – a combination of green and grey options

Hybrid approaches combine the construction of specific grey options or built infrastructure with the simultaneous installation of restored or newly created natural infrastructure. For example, removable sea walls or flexible flood gates can be installed simultaneously with salt marsh and oyster reef restoration. Combining green or blue and grey protect measures is expected to be more effective and less costly under particular circumstances (Browder et al., 2019). For example, a hybrid approach can be implemented whereby natural infrastructure provides protection benefits for small to medium events, while built infrastructure is included in the measure for additional protection against larger events. Advantages of the hybrid approach include that it can be used in areas where there is little space to implement natural measures alone, it capitalises on the best characteristics of built and natural measures, it allows for innovation in designing coastal protection systems, and it can provide a greater level of confidence than natural approaches alone (Sutton-Grier et al., 2015).

**Case study – coastal lagoon of Aveiro, Portugal**

The coastal lagoon of Aveiro, Portugal, has long been studied for its peculiar configuration, high biodiversity and ecological value and its severe exposure to natural hazards (Lopes et al., 2017; Mendes et al., 2021; Pinto et al., 2020; Ribeiro et al., 2021; Stronkhorst et al., 2018). Situated along the Atlantic coast, Aveiro is extremely vulnerable to coastal erosion and SLR and thus requires integrated and sustainable management of coastal resources. Accordingly, over the last decade, Aveiro has applied a hybrid approach to coastal management by combining adaptation measures that mix traditional hydraulic engineering with green options (Stronkhorst et al., 2018), also known as "building with nature" (Chen et al., 2022).

One of the distinguishing aspects used in Ria de Aveiro is the combination of hard defences, beach nourishment and restoration of wetlands. Over the years, Aveiro has built approximately 10 sea walls and 20 groynes and combined these hard defences with beach nourishment along the coast to reinforce and enlarge beaches, providing natural barriers against tides and storms (Stronkhorst et al., 2018). Along with the latter two measures, Aveiro has restored previously abandoned salt pans. The latter plays a fundamental role in the mitigation of flooding and the protection of coastal communities as it increases the capacity to absorb excessive water during high tides and storm surges, thereby creating a natural protection against flooding. Overall, the hybrid approach has helped to increase the resilience to climate change in the coastal area of Aveiro, protect local communities, enhance recreational use and finally preserve coastal ecosystems.

Box 3: Sea level rise and World Heritage Sites: the case of the Wadden Sea

SLR and associated coastal hazards have been identified as a major threat to both natural and cultural coastal world heritage (Marzeion and Levermann, 2014; Sesana et al., 2020). Recent studies indicate that accelerating SLR is expected to exacerbate the pressure on World Heritage Sites (WHSs) through, among others, more frequent flooding or increasing erosion, with the number of threatened sites increasing sharply towards the end of the century in all scenarios (Reimann et al., 2018; Vousdoukas et al., 2022). For cultural heritage, potential impacts may range from direct damage to archaeological structures, buildings and monuments to changes in landscapes and visitor behaviour (Phillips, 2015). For natural WHSs, coastal erosion, permanent submergence and salt intrusion are examples of SLR-related processes that may alter the character and nature of a site, thus affecting its Outstanding Universal Value.

Adaptation of WHSs to SLR is particularly complex due to the potentially adverse implications of adaptive measures for heritage significance (Phillips, 2015) but also because different sites, due to their nature, have very different adaptation needs and no "one-fits-all solution" exists. Nevertheless, in some cases, natural areas may accommodate some of these disruptions and maintain ecological equilibrium by migrating landwards (Vousdoukas et al., 2022), if not constrained by coastal development, or even seawards where conditions allow. However, little information exists in the literature regarding potential adaptation options for heritage managers and policy-makers (Reimann et al., 2018). Although some adaptation options such as managed retreat, ecosystem-based adaptation and relocation have been proposed in the context of WHS adaptation to SLR (e.g. Vousdoukas et al., 2022), which mainly due to their non-intrusive nature appear to offer promising alternatives in some cases, a better understanding regarding their effectiveness and their suitability for specific sites is required for their implementation. Further adaptation barriers include the lack of institutional frameworks and policies specific to WHSs as well as financial and socio-cultural barriers (Fatorić and Biesbroek, 2020).

One example of adaptation of WHSs comes from the Wadden Sea, which has been a UNESCO World Heritage Site since 2009. The Wadden Sea is located in the North Sea between the coastlines of Denmark, Germany and the Netherlands and is the largest unbroken system of intertidal sand and mud flats in the world and one of the last remaining large-scale, intertidal ecosystems where natural processes continue to function largely undisturbed. The site includes the Dutch Wadden Sea Conservation Area, the German Wadden Sea National Parks of Lower Saxony and Schleswig-Holstein and a large part of the Danish Wadden Sea maritime conservation area (UNESCO, 2023). It is a large coastal wetland environment with tidal channels, sandy shoals, sea-grass meadows, mussel beds, sandbars, mudflats, salt marshes, estuaries, beaches and dunes (Schuerch et al., 2014; UNESCO, 2023), the development of which is driven by diverse morpho- and hydro-dynamics (Benninghoff and Winter, 2019). SLR projections for the Dutch Wadden Sea show a significant rise for all the scenarios and, in particular, a rise of $0.76 \pm 0.36$ cm under representative concentration pathway (RCP) 8.5 (Vermeersen et al., 2018).

Accelerated SLR can have important implications for the Wadden Sea, affecting sediment balance and potentially leading to permanent submergence in parts, despite its intertidal flats being effective sediment sinks and appearing to be quite resilient against even high rates of SLR (Hofstede et al., 2018). In fact, data from the last 2 decades indicate an expansion of intertidal areas but a reduction and deepening of subtidal areas and channels in some parts (Benninghoff and Winter, 2019). However, observed changes in tidal asymmetry in the German Wadden Sea suggest that sediment accretion trends may be coming to an end (Hagen et al., 2022). Furthermore, future projections indicate a transition from a tidal-flat-dominated system to a lagoon-like system, despite increased accumulation of sediment in the back-barrier basin, as this accumulation appears to be far too weak to compensate for the rise in mean sea level (Becherer et al., 2018). Such changes can potentially have dramatic implications for the unique ecosystem of the Wadden Sea (Becherer et al., 2018). Moreover, beyond a critical rate of SLR, major changes in ecotope distribution are projected to occur (Timmerman et al., 2021), and adaptation strategies such as inland migration of the shoreline can result in larger impacts, including the formation of a deep tidal basin with large subtidal habitats and a shifted intertidal zone (Timmerman et al., 2021). Besides SLR, potential changes in storm activity and characteristics can further affect the development of the site, particularly its wetlands, partially exacerbating or even counteracting the effects of SLR (Schuerch et al., 2013).

Although the future of the Wadden Sea under SLR appears to be a topic of concern and the need for adaptation is widely recognised (e.g. Heron et al., 2020), little has been done in terms of developing adaptation plans for the region. This is, in part, due to complexities related to the nature of the site, existing coastal protect measures and the involvement of three countries in its management. An example of such a plan is the integrated climate change adaptation strategy established by the German state of Schleswig-Holstein with the aim of maintaining the present functions and structures as well as the integrity and dynamic nature of the Wadden Sea ecosystem over the long term for its section of the Wadden Sea site (Hofstede and Stock, 2018). Developing such plans for the entire basin presents many challenges but is imperative for preserving the Wadden Sea and maintaining its World Heritage status.

# 3 Approaches for decision-making

This section presents approaches suitable for supporting coastal adaptation decision-making. A large number of approaches (methods, tools) are available in the literature and are being applied in practice to support coastal adaptation decisions (i.e. to find a suitable alternative given some criteria), and it is impossible to provide a comprehensive overview. Hence, we limit ourselves here to presenting key aspects that need to be considered in coastal adaptation decision-making, together with some example tools that can be used for addressing them. Towards this end, we first clarify the decision science terminology (Sect. 3.1) and review the common characteristics of coastal adaptation decisions (Sect. 3.2). Then, the section continues to present the key aspects that need to be considered in coastal adaptation decision-making, which are (i) considering multiple criteria and interests (Sect. 3.3), (ii) implementing low-regret and flexible options (Sect. 3.4), (iii) keeping future options open (Sect. 3.5), (iv) factoring SLR into decisions that need to be made today (Sect. 3.6) and (v) revisiting decisions iteratively together with monitoring (Sect. 3.7).

## 3.1 Decision science terminology

A decision involves a pre-defined set of options (also called alternatives or actions) to choose from, wherein each alternative can consist of a combination of measures. For example, common coastal adaptation measures include upgrading dikes, restoring coastal wetlands and installing building-level flood shields. An adaptation option may then consist in increasing the dike height by 1 m, restoring salt marshes in front of the dike and implementing flood shields to protect against floods with a water depth of 2 m. Typically, coastal decisions are not one-shot decisions but consist of sequences of decisions over time. Hence, the decision consists in choosing an adaptation pathway, which is a sequence of options applied over time (also called "policy" or "strategy" in some branches of decision science). Note that this general notion of adaptation pathways is independent of the method "adaptation pathway analysis" (Haasnoot et al., 2013), which is one tool that can be applied to produce adaptation pathways.

Approaches (methods, tools) to decision-making involve both participatory and analytical methods, which fulfil complementary roles in supporting adaptation decisions. Participatory methods (also called transdisciplinary, co-production or co-creation methods) target the social processes of learning and cooperating among stakeholders and possibly researchers (Anderson and McLachlan, 2016; Cornwall, 2008; Watson, 2014). Analytical methods, in turn, support the identification of suitable options or adaptation pathways in those situations in which it is not obvious what to do. They do so by helping to identify options that perform best or well with regards to the preferences of the stakeholders. Towards this end, each option is characterised by one or several criteria, which measure any relevant social, ecological or economic value associated with choosing and implementing the alternative (Kleindorfer et al., 1993). Criteria commonly used in the coastal adaptation domain include cost of options, avoided damages, longevity of options, robustness of options, flexibility of options as well as social acceptance.

## 3.2 Common characteristics of coastal adaptation decisions

Coastal adaptation decision-making is challenging due to the following characteristics.

- *Diversity of fundamentally different measures.* Section 2 highlighted that there are four fundamentally different ways to respond to SLR (protect, accommodate, advance and retreat), with each way having advantages and disadvantages. In addition, each of these categories entails many measures, which again come with their own advantages and disadvantages.

- *Multiple objectives and trade-offs.* Whatever approach to coastal adaptation is taken, the choice and planning of adaptation pathways generally need to consider multiple objectives. Adaptation policy is not only about SLR and flood risk but also needs to consider many other policy objectives, such as socio-economic development, human safety, biodiversity and water quality as well as the numerous human activities that coastal systems support, including shipping, agriculture, aquaculture, tourism and fishing. Therefore, there is generally no single "best" solution that satisfies all objectives. Instead, coastal adaptation decisions are characterised by trade-offs. For example, restoring wetlands for coastal protection and biodiversity reduces the space available for industrial or urban land use.

- *Diverse interests and social conflict.* Coastal decisions are generally characterised not only by multiple objectives, but also by diverse and often conflicting interests of stakeholders involved in and affected by the decisions, which gives rise to social conflicts (Oppenheimer et al., 2019). For example, homeowners or tourism operators may prefer not to have dikes in front of their homes if these jeopardise the view of the beach. As a consequence, stakeholders generally disagree on how to rank objectives or which criteria to apply for measuring progress towards objectives (see Bisaro et al., 2024, for governance arrangements, e.g. Marine Spatial Planning to address diverse interests in coastal adaptation).

- *Long-time horizons.* Many coastal decisions involve adaptation measures with long lead times and lifetimes (Haasnoot et al., 2020). For example, coastal protection infrastructure such as dikes, sea walls and breakwaters usually involves decision horizons of 30 to 100 years

and more (Burcharth et al., 2014), and major protection infrastructure such as storm surge barriers generally takes decades to plan and implement and hence may be built for even longer lifetimes (Gilbert and Horner, 1986). Similarly, land use planning, coastal risk zoning and coastal realignment decisions (Hino et al., 2017) may have effects that last several decades, extending to over a century.

- *Large and deep uncertainties.* The long-time horizons involved in some coastal adaptation decisions are specifically challenging due to the large and deep uncertainties involved in long-term projections (i.e. 50 years and more) of SLR. Deep uncertainty means that SLR experts cannot attach a single unambiguous probability distribution to future SLR, because they cannot agree on an unambiguous method for deriving probabilities or because their subjective probability judgements differ (Kwakkel et al., 2010; Lempert and Schlesinger, 2001; Weaver et al., 2013). Projections of long-term SLR and other climate change variables are generally deep, because these depend on emission scenarios. However, also within a given emission scenario, uncertainty is large. For example, according to the latest IPCC report, there is a 65 % chance that sea levels will rise by 0.6 to 1.0 m until 2100 in all emission scenarios considered, with increases of up to 1.6 m or more also being possible (Fox-Kemper et al., 2021).

## 3.3   Considering multiple criteria and interests

Given the multi-objective and social conflict nature of the coastal decisions described above, participatory methods and multi-criteria decision analysis (MCA) methods can support most coastal decisions. MCA methods are standard methods for addressing multi-objective problems. These methods help stakeholders to structure the process of decision-making into a series of steps, to identify their preferences and to choose an option that is consistent with those preferences (Cinelli et al., 2020; Greco et al., 2016). For example, the MCA method called analytical hierarchy process guides stakeholders through pairwise comparisons of criteria in order to transform their preferences into weights for aggregating criteria into a single score for each option (Saaty, 1980). MCA methods have been applied widely in a coastal context (Townend et al., 2021; Le Cozannet et al., 2013; Hinkel et al., 2023). These methods are also an integral part of many decision-making tools, such as dynamic adaptation policy pathway (DAPP) analysis (Haasnoot et al., 2013), to which we will return later below.

MCA methods can, to some extent, also contribute to addressing social conflicts, e.g. by supporting the analytic search for compromises between stakeholders' divergent preferences (Munda, 2008), but the suite of available participatory methods entails much more, also beyond those methods that have a more analytical focus. Examples of such approaches include climate risk narratives (Jack et al., 2020), anticipatory learning (Tschakert and Dietrich, 2010), living laboratories (Bergvall-Kåreborn and Ståhlbröst, 2009) and citizens' juries, planning cells and consensus conferences (Escobar and Elstub, 2017). Generally, the normative literature on adaptation suggests that any analytical method for supporting adaptation should be embedded in a participatory process that includes all stakeholders in order to build trust, enhance legitimacy, reduce social conflicts and advance fairness and justice (Michels and De Graaf, 2010; Callahan, 2007; Irvin and Stansbury, 2004).

It is important to note that participation is not automatically a key to success. A growing empirical literature that describes how adaptation processes play out in practice shows that participatory processes often fail to deliver, either because they are poorly designed and implemented, conflicts cannot be overcome, or interests of powerful actors dominate outcomes (Harman et al., 2013; Oppenheimer et al., 2019). This resonates with a larger empirical literature in the field of public participation, which has found that many participatory processes are tokenisms, in which the have-nots are informed or heard but the power-holders retain the right to decide (Hoppe, 2011; White, 1996; Arnstein, 1969).

Two conclusions can be drawn from this discrepancy between the normative and descriptive literature. First, more empirical work is needed for understanding under which conditions participatory adaptation processes deliver. Second, it needs to be acknowledged that participation cannot solve all problems, in particular not those related to power asymmetries rooted deeply in social structure.

## 3.4   Implementation of low-regret measures

One immediate and generally recognised priority in coastal adaptation is the implementation of no- or low-regret measures. What this means in practice depends on the context, but generally this includes generic accommodate measures such as awareness raising, emergency planning and early-warning systems (Lumbroso et al., 2017). The strength of these measures is that they have high cost–benefit ratios over short time horizons, which means that implementing them today produces almost immediate net benefits (Oppenheimer et al., 2019). Early-warning systems have one of the highest cost–benefit ratios and should be a universal response (Rogers and Tsirkunov, 2010). However, these measures alone are only effective for current conditions, and small rises in sea level therefore need to be combined and/or replaced with other approaches if SLR is substantial.

Other low-regret measures can be found when addressing the local drivers of relative SLR and coastal hazards. These may include (1) the preservation of coastal wetlands to reduce both surge and wave impacts as well as the maintenance of sufficient accommodation space for these to migrate inland with SLR; (2) the maintenance of natural sediment sup-

ply by reducing dam building in rivers, which in turn reduces the risk of wetland loss and erosion; and (3) the reduction of anthropogenic drivers of subsidence and building land elevation with natural processes (Nicholls et al., 2021b).

Retreat is generally not a low-regret measure for densely populated and heavily used coastal areas, but it may be for rural areas if sufficient space is available to convert dry land into coastal wetlands that contributes to coastal protection. In the aftermath of disaster, retreat may also become low-regret for more densely populated zones when reconstructing livelihoods in situ becomes as costly as relocating. After Superstorm Sandy, for example, a number of flooded formerly developed areas around New York were purchased and not rebuilt, although this was a reactive rather than proactive response (Braamskamp and Penning-Rowsell, 2018). In Europe, one example of retreat happening after a disaster was Cyclone Xynthia, which hit the French Atlantic coast in February 2010, killing 47 people and causing total damages of about EUR 1.5 billion, which led to the decision to relocate some houses and neighbourhoods (Rouhaud and Vanderlinden, 2022). It must, however, be noted that part of this decision was later taken back due to strong civil opposition, which illustrates the difficult and socially contested nature of coastal retreat in general (Hino et al., 2017).

## 3.5 Keeping future options open

Given the large uncertainty about by how much sea levels will rise in the coming decades, an important policy priority is to keep future options open (Hinkel et al., 2019; Hallegatte, 2009). One way to do this is to postpone long-term decisions that do not need to be made today. Many decisions about retreating from the shoreline, in particular for urban areas, fall into this category (Oppenheimer et al., 2019). While SLR may rise by several metres, posing existential threats to coastal zones, there is also a substantial chance that SLR may stay below 30 cm by 2100 (50th percentile of SPP1–1.9) if Paris Agreement goals are reached. Protecting coasts from the latter amount of SLR is economically efficient and relatively cheap for about 90 % of the global population, as coastal population tends to be concentrated in coastal urban areas making up about 10 % of the global coastline (Lincke and Hinkel, 2018; Tiggeloven et al., 2020; Vousdoukas et al., 2020). Hence, a practical strategy for urban areas is to wait and observe how SLR observations and projections develop over the next decades, providing a robust basis for retreat versus protect decisions (Hinkel et al., 2019).

Another way of keeping future options open is by implementing flexible options that can be upgraded or changed over time once more is known about future SLR. This is generally an argument in favour of implementing soft and sediment-based measures such as NbS instead of hard measures, because the former can either self-adjust to relative SLR (in the case of coastal wetlands; see Box 2) or can easily be adjusted (in the case of sediment nourishment). However,

flexibility can also be built into hard infrastructure. For example, in Germany, new coastal dikes are built with a wider crest than is necessary today, which allows further raising at low costs if SLR turns out to be higher than originally anticipated (MELUR-SH, 2012).

Postponing the decision and building flexibility in the current options raises questions of timing: by how much a decision should be postponed or how much flexibility should be built in. These questions can be addressed from an economic point of view by a class of methods termed real-option analysis (ROA), which is covered in the next subsection.

## 3.6 Factoring SLR into decisions that need to be made today

Some long-term decisions cannot be postponed and need to be made today. This may include decisions related to critical infrastructure, urban renewal, inadequate coastal protection, land use planning and land reclamation. As these and similar decisions have time horizons of decades to over a century (Azevedo de Almeida and Mostafavi, 2016; Haasnoot et al., 2020), factoring SLR into such decisions is beneficial. A range of analytical methods for supporting these kinds of decisions exists.

One classical set of methods for decision-making under deep uncertainty (i.e. without probabilities) is *robust decision-making* (van der Pol et al., 2023), which refers to a range of methods that identify adaptation measures that are effective in a wide range of scenarios (Heal and Millner, 2014; Lempert and Schlesinger, 2001; Wilby and Dessai, 2010). This includes so-called exploratory modelling, which uses models to create a large ensemble of plausible future scenarios and then searches visualisation techniques to identify robust options (Lempert and Schlesinger, 2000). Robust decision-making (RDM) also includes methods that follow similar ideas, such as robust optimisation (Ben-Tal et al., 2009), information gap theory (Ben-Haim, 2006) and classical approaches such as minimax and minimax regret (Savage, 1951). The latter approaches (i.e. minimax or minimax regret) are simple and low burden to apply and constitute a useful addition to e.g. standard cost–benefit analysis carried out for different sea level rise scenarios (van der Pol et al., 2021). The more complex approaches such as exploratory modelling and robust optimisation are generally applied in the context of an expensive coastal infrastructure project, such as upgrading the port of Los Angeles? (Sriver et al., 2018).

Another set of analytical methods for long-term decision-making under SLR is found in the so-called *adaptive decision-making* methods. These methods are suitable if adaptation decisions are not made as single-shot decisions today but as sequences of decisions at several moments in time, a situation frequently found in the coastal adaptation context. These methods aim at finding adaptation measures that are robust against a wide range of futures in that they are

flexible to allow adjustments over time once more about SLR is known (New et al., 2022; Marchau et al., 2019).

Broadly, two categories of analytical adaptive decision-making (ADM) approaches exist (Völz and Hinkel, 2023). A first category of these methods starts with a user-defined set of adaptation options and then an analysis of how these options can be sequenced over time in different scenarios (e.g. SLR) in order to achieve the desired objectives (Walker et al., 2001). A widely used tool for such adaptive planning is adaptation pathway analysis (Haasnoot et al., 2013, 2012), which graphically explores how available adaptation measures can be sequenced over time, in order to reach adaptation goals. This analysis also considers the lead times of adaptation measures (i.e. the time needed for planning and implementing adaptation measures), because rapid SLR may lead to insufficient time being left to plan and implement measures with long lead times, such as surge barriers, as these usually take decades to plan and implement (Haasnoot et al., 2020). A prominent example where this approach has been applied is the Thames Barrier in the UK, which protects the city of London. Within the Thames Estuary 2100 project, adaptation pathway analysis has been applied, next to other approaches, in order to find out whether there is sufficient time to upgrade or replace the Thames Barrier under a rapid acceleration of SLR (Ranger et al., 2013).

The second category consists of *economic ADM approaches*, which identify optimal adaptation decision rules by taking into account information about what will be learned in the future about the development of key climate variables. These methods are often found under the labels of real-option analysis (Wreford et al., 2020) and optimal control studies (Hermans et al., 2020). Importantly, these methods consider future learning about relevant variables (e.g. mean and extreme sea levels) in the economic valuation of adaptation measures in order to find optimal trade-offs between investing today, including the cost of flexible design, and postponing investment decisions until additional information is available (Dixit and Pindyck, 1994). Hence, these methods can provide justifications for whether implementing flexible adaptation measures today are worth the extra costs. This is specifically relevant for public decisions that involve expensive and long-lasting infrastructure, as found on coasts, because the public sector needs to justify public money being spent wisely. While ROA applications of adaptation to coastal and river floods are growing (Dawson et al., 2018; Kim et al., 2019; Hino and Hall, 2017; Linquiti and Vonortas, 2012; Woodward et al., 2011, 2014; Ryu et al., 2018), to date they are poorly connected to state-of-the-art SLR science. The first steps towards closing this gap were taken by Völz and Hinkel (2023), who developed SLR learning scenarios based on the SLR scenarios of the IPCC's Sixth Assessment Report (AR6).

A critical and difficult decision that needs to be made in the application of all of the above-mentioned decision analysis methods is how much SLR should be considered in a particular decision. Importantly, sea level science can only give a partial answer to this question, because the other part of the answer depends on the uncertainty preferences of the stakeholders involved in and affected by the decisions. When stakeholders are uncertainty-tolerant and the value at risk is relatively low, then the "standard" IPCC scenarios, which provide a so-called *likely* range of possible future SLR, are a good basis for decision-making (Oppenheimer et al., 2019). If stakeholders are less tolerant of uncertainties, which is often the case in urban contexts, then higher SLR scenarios should also be considered. This is because the IPCC's *likely* range is the 66 % central interval of future SLR, which means there is a 17 % chance of SLR exceeding the likely range, which may be too large a chance for uncertainty-averse stakeholders (Hinkel et al., 2015; Nicholls et al., 2021a). In this case, more unlikely SLR scenarios should be considered, with the exact choice depending on the stakeholders in the specific case. The IPCC AR6, for example, states that, in the case of unlikely but rapid melting of the ice sheets, a 2 m rise in sea level by 2100 cannot be excluded in an unabated emission scenario (SPP5–8.5) (Fox-Kemper et al., 2021).

## 3.7   Revisiting decisions iteratively and monitoring

No matter which decision analytical method is applied, a final and critical priority is to set up an iterative policy- and decision-making process (Fig. 9) that regularly revisits decisions and that includes a monitoring framework, through which SLR and other relevant variables are monitored and appropriate action can be triggered if a relevant threshold is crossed (Walker et al., 2001, 2013). The idea is to implement no- or low-regret options and flexible measures today and then monitor SLR, ESL and other decision-relevant variables in order to be able to identify when decisions and new policies are required. Importantly, a monitoring system is essential for identifying the need for action in sufficiently early time to allow planning and implementation before negative impacts occur (Hermans et al., 2017). One well-known framework that entails this idea (and combines it with the adaptation pathway analysis covered in the last subsection) is DAPP (Haasnoot et al., 2013). This method has been widely applied in various contexts and has, for example, been integrated into the national guidance for coastal hazard and climate change decision-making in New Zealand (Lawrence et al., 2018).

## 4   Summary: key developments per basin

Adaptation to SLR in Europe has been approached through various types of measures to accommodate, protect, advance and retreat. Adaptation strategies on Europe's coasts thus constitute a mix of hard and soft measures, planning measures, policy developments and stakeholder and community engagements. Below, we summarise the main developments organised by the different sea basins.

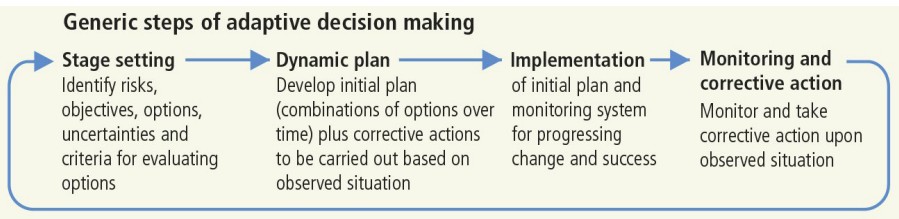

**Figure 9.** The adaptive decision-making cycle. Source: extracted from the original figure available at SPM.5d (IPCC, 2019).

In the Baltic Sea basin, for accommodate measures, progress has been made, with several Baltic nations incorporating SLR projections into their spatial planning and land use regulations. Notably, Estonia has implemented a Maritime Spatial Plan for 2022 that integrates SLR information. In terms of protect measures, upgrading coastal defences, e.g. with sea walls, embankments and dikes, has been implemented, while nature-based solution initiatives to restore and create wetlands and coastal marshes that can act as buffer zones and reduce wave energy are also underway. For instance, the Danish Baltic coast provides the first large-scale example of successful managed realignment with the restored Gyldensteen Coastal Lagoon, which has to date enhanced ecological status and species richness in the project area (Thorsen et al., 2021). The Baltic Sea basin has also seen progress in marine environment conservation, which can potentially enhance living marine resources and related fishing activities. Key to furthering coastal adaptation in the basin is ensuring that solutions are also linked to financing mechanisms that can mobilise co-finance, e.g. from the private sector, to supplement national public funding.

In the North Sea basin, SLR information has been integrated into coastal planning at the national and sub-national levels in most countries, while North Sea basin countries are implementing different mixes of hard and soft protect measures. In the Netherlands, the Delta Programme includes a comprehensive mix of measures to maintain a healthy groundwater system, using spatial planning and other context-specific strategies while providing more space for water and enhancing urban and ecological values. Sand nourishment is also growing in importance as a coastal protect measure in the Netherlands, alongside dike upgrading and reinforcement. In Germany, there is an emphasis on integrated coastal zone management and dike upgrading and widening that incorporates flexibility for future SLR. In the UK, a mix of protection, beach nourishment and managed retreat is being considered for different portions of the coastline. These countries each reflect different approaches to addressing uncertainty that should be iterated and revisited as more information on SLR becomes available in the future.

In the Mediterranean Sea basin, key developments include the mainstreaming of SLR information into planning through the development of national adaptation plans, e.g. in Spain and Italy. Furthermore, insurance is emerging as an accommodate measure to address SLR-related risks, e.g. in Spain and France. Soft protect measures, such as sand nourishment and nature-based solutions more broadly, are important in the Mediterranean Sea basin, with coastal reforestation and the restoration of dunes and marshes implemented in various regions to act as natural barriers. Other examples are cliff strengthening and stabilisation measures that include green and grey options focusing on reducing erosion and enhancing natural protection along coastal cliffs, e.g. in Croatia and Italy. Several major urban areas in the basin have initiated large-scale adaptation measures. For example, the Venice MOSE project is a system of mobile barriers constructed to protect Venice from high tides and flooding, while the city of Barcelona has introduced green infrastructure projects that focus on permeability and water retention to combat both SLR and increased rainfall. Such differentiated measures appropriate to the specific biophysical and socio-economic context at issue should be further supported through participatory co-development approaches for coastal decision-making (Bisaro et al., 2024).

In the Black Sea basin, there is an increased emphasis on developing monitoring and early-warning systems to help manage SLR and the associated flood risks. Furthermore, efforts have focused on upgrading and modernising existing coastal infrastructure to enhance resilience to rising sea levels. For example, in Romania, a major initiative combining sand nourishment and cliff stabilisation with marine measures including artificial reef building is being implemented to reduce coastal erosion risks exacerbated by SLR and to enhance resilience in the tourism sector. Furthermore, implementation of such nature-based solutions that also benefit local economies is promising and should be explored for scaling up coastal adaptation in the basin.

In the Atlantic Ocean basin, countries are implementing a range of adaptation measures, with an emerging focus on nature-based solutions and improved spatial planning to reduce risks to coastal development across the entire basin. Soft protect measures, such as cliff strengthening and sand nourishment, are being implemented in Portugal, while restoration measures, protecting against wave energy and therefore limiting erosion and sediment accumulation, are being implemented in Spain, Portugal and France. Advance strategies are also being implemented through nature-based solution approaches, as in Spain, where the national

adaptation plan envisions the regeneration of beaches and artificial dune systems to reduce erosion and revitalise coastal ecosystems, e.g. in the restoration one of the largest dune systems of the Cantabrian Sea. Furthermore, in France, coastal land in the south-west of the country has been advanced with the creation of a vegetated area with the specific intention of supporting natural accretion of land and surrounding low areas. Finally, retreat measures are also being implemented, as in Portugal, where the progressive removal of constructions located in flood-critical territories along the coastline is being implemented through spatial planning instruments to manage the risk of SLR.

Across all the basins, a common theme is the shift towards a combination of traditional engineering solutions with soft measures, including nature-based solutions. Integrating local communities into decision-making processes and emphasising the importance of continuous monitoring and flexible management strategies, e.g. through coastal planning instruments such as Marine Spatial Plans (Bisaro et al., 2024) and the other adaptation decision-making methods discussed above, are also notable trends. Ensuring that these trends lead to appropriate mixes of coastal adaptation measures being found depends on the continued support and involvement of public and private sector stakeholders in effective multi-level governance.

## 5 Conclusions

This paper has conducted a review of the literature on coastal adaptation and analysed 17 adaptation measures targeting climate impacts, such as coastal flooding, saltwater intrusion, coastal erosion and impacts on ecosystems and estuaries. Some examples of coastal adaptation measures that have been discussed are early-warning systems, insurance and policy instruments, hard and soft defences, nature-based adaptation measures, newly raised ports and planned relocation. At the sea basin level, Baltic countries are incorporating SLR projections into their spatial planning and land use regulations, and progress has also been made in marine environment conservation. In the North Sea basin, SLR information has been integrated into coastal planning at national and sub-national levels in most countries, and countries are implementing different mixes of hard and soft protect measures. In the Mediterranean Sea basin, SLR information is being mainstreamed through the development of national adaptation plans. Prominent protect measures are coastal reforestation and dune and marsh restoration, while insurance is emerging as an accommodate measure. In the Black Sea basin, emphasis is on early-warning systems and on upgrading and modernising existing coastal infrastructure to enhance resilience. In the Atlantic Ocean basin, an emerging focus of adaptation measures is on nature-based solutions and improved spatial planning. In addition, the measures discussed in this paper are generally subject to trade-offs that

should be considered when planning for coastal adaptation. In order to accurately analyse existing trade-offs, it is important to understand the effectiveness and feasibility of these measures. Future research can expand the literature review to include more studies, and more research is needed to learn about the trade-offs of implementing each of these measures as well. The approaches for decision-making showed that coastal adaptation is a complex undertaking, given the large number of possible and diverse adaptation measures available as well as the equally large set of participatory and analytical methods available for supporting this process. Furthermore, context and decisions to be made, as well as experience in coastal adaptation, differ significantly from place to place and from region to region across Europe. Whereas northern Europe and also some parts of southern Europe such as the Po Delta have been protected against the sea for decades to centuries and have long experience in adapting to relative SLR, for most of southern Europe, coastal adaptation is a new necessity. In both contexts, decisions differ in terms of the time horizons considered, the sizes of the investments involved as well as the preferences decision-makers and their constituencies have for accepting risk. For all of these diverse situations, analytical tools are available to support decision-making, ranging from relatively low-burden tools such as adaptation pathway analysis and multi-criteria analysis to technically sophisticated methods such as robust decision-making and real-option analysis. Regarding the participatory approaches for supporting decisions, which were not the focus of this paper, it can be concluded that there is a large discrepancy between the normative and descriptive literature: while there are many papers and guidelines available recommending what there is to do, the empirical evidence on whether this works in practice is relatively thin. Hence, more empirical work is needed for understanding under which conditions participatory adaptation processes deliver. However, even if we learn more about what works and what does not work in practice, it needs to be acknowledged that participatory methods cannot solve all problems, in particular not those related to power asymmetries rooted deeply in society.

**Data availability.** No data sets were used in this article.

**Author contributions.** GG, EFB and AB acted as coordinating lead authors. GG, JH, EFB, AB and RBA wrote the paper with textual contributions from OE and MFC. ATV wrote the box of the Wadden Sea and PC the box of the MOSE system in Venice. All the authors participated in the iterations and revisions of the paper.

**Competing interests.** The contact author has declared that none of the authors has any competing interests.

**Disclaimer.** Publisher's note: Copernicus Publications remains neutral with regard to jurisdictional claims made in the text, published maps, institutional affiliations, or any other geographical representation in this paper. While Copernicus Publications makes every effort to include appropriate place names, the final responsibility lies with the authors.

**Acknowledgements.** We would like to acknowledge the guidance of Nadia Pinardi that inspired the idea of this chapter.

**Review statement.** This paper was edited by Bart van den Hurk and reviewed by Pablo Fraile-Jurado and one anonymous referee.

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
