# Peer review of "Sea Level Rise in Europe: Adaptation Measures and Decision Making Principles"

_State of the Planet, 2023_

## Community Comment (CC1)

**Daria Povh Škugor and Ivan Sekovski**
**PAP/RAC**

**General comment:**

For an article that aims - to provide guidance for the design and implementation of adaptation policies in European basins, the question is posed: why is existing legislation not considered? Also, strategies and plans prepared for coastal management and adaptation are missing.

As for the legislation, the question is primarily related to the ICZM Protocol of the Barcelona Convention, which is the only regional legal text in the world inviting countries to establish a setback zone (Article 8) – a low-regret measure for the rising sea level. EU has ratified the ICZM Protocol, and by that ratification, the Protocol became part of European Union law.
https://eur-lex.europa.eu/legal-content/EN/TXT/?uri=celex%3A22009A0204%2801%29

In addition, the Protocol, in its article 18, invites Mediterranean countries to prepare national ICZM Strategies, coastal plans and programmes. All strategies, plans and programs developed since the Protocol entered into force were focused on adaptation to the changing climate within its commonly applied integrated approach. Within the extensive scientific and grey literature reviewed (including 206 items), none of the UNEP/MAP, PAP/RAC documents have been considered. Although ClimateAdapt is considered, AdriAdapt is not.

**Comments per lines**

Lines 105 and 600 Setback has not been included as the measure, although it is among the most effective low-regret measures in general.

Lines 100-120 and 185

One of the accommodation measures is the establishment of a setback zone. This measure is requested by the Protocol on ICZM in the Mediterranean, the world's first regional legislation for coastal management. Article 8 of the Protocol on "*Protection and sustainable use of the coastal zone*" invites Mediterranean countries to limit new constructions and the coastal linear extension of urban developments and transportation through the provision of coastal setback zones[1],[2], [3]. The delimitation of these zones must be at least 100 m in width, taking into account the highest winter waterline and the areas directly and negatively affected by climate change and natural risks. Several EU countries have setback regulations within their national coastal laws (e.g., France and Spain), and some within spatial planning law (e.g., Croatia). In Italy, the coastal setback zone, as a method for coastal risk reduction, is proposed as an effective integrated strategy for future coastal planning in the Italian region of Emilia-Romagna[4],[5]. Coastal setback zones provide coastal protection by reducing
* * *
[1] UNEP/MAP/PAP. (2008). Protocol on Integrated Coastal Zone Management (ICZM) in the Mediterranean. Split, Priority Actions Programme.

[2] PAP/RAC. (2021). *Coastal Resilience Handbook for the Adriatic.* INTERREG AdriAdapt project, Split. https://adriadapt.eu/wp-content/uploads/2022/01/Coastal-Resilience-Handbook-for-the-Adriatic.pdf

[3] Ocean & Climate Platform. (2022). Adapting Coastal Cities and Territories to Sea Level Rise in the Mediterranean Region: Challenges and Best Practices. Ocean & Climate Platforme. 48 pp.

[4] Perini, L., Calabrese, L., Salerno, G., Ciavola, P., and Armaroli, C. (2016) Evaluation of coastal vulnerability to flooding: comparison of two different methodologies adopted by the Emilia- Romagna region (Italy), Nat. Hazards Earth Syst. Sci., 16, 181– 194, https://doi.org/10.5194/nhess-16-181-2016

[5] Emilia Romagna Region. (2022). Strategia di Gestione Integrata della Costa ai cambiamenti climatici (GIDAC). INTERREG AdriaClim project. https://ambiente.regione.emilia-romagna.it/it/suolo-bacino/argomenti/difesa-della-costa/gidac/gidac-dicembre-2022/strategiagidac_documento_dic22.pdf

the number of assets (e.g. houses, infrastructure, and businesses) in areas susceptible to coastal hazards, which are expected to increase with climate change. In other words, they provide a buffer to coastal flooding and erosion.[6] Coastal setback zones ensure open public spaces and access to the shoreline which increases the opportunity for the development of tourism, beach economy and recreational activities. Through the prioritisation of public services and activities, they improve the quality of coastal experiences among residents and tourists[2]. Implementation of coastal setback zones secures space for ecosystems and for the creation of Nature-based Solutions. As such, they may also attenuate wave action. There are other benefits of implementing coastal setback zones related to additional ecosystem services, such as maintaining the water quality and allowing erosion and accretion cycles to occur naturally, thus retaining sediment budgets. Additionally, setback zones can have a multifunctional role, and be a part of the EU policy that promotes the use of nature-based green and blue infrastructure, improving environmental conditions and mitigating negative effects of the built environments in cities.[6]

Finally, by limiting urban development in the short term, setback zones facilitate realignment in the long run. Studies have demonstrated the effectiveness of these setback zones in combination with coastal adaptation measures (i.e. managed retreat and protection) to reduce present and future costs of coastal flooding[7]. As regards the future sea levels, the size of buffer zones between structures and the sea will inevitably reduce, thereby compelling the periodical review of their delimitations to ensure they continue to provide sufficient protection[3]. This measure is increasingly being adopted among Mediterranean countries despite delays and exceptions which have permitted different developments[8].

8        Integration of Climate change adaptation in coastal zone management plans and strategies
Integrated Coastal Zone Management Plan of the Šibenik-Knin County, Croatia[9]
Coastal Zone Management Plan for the City of Kaštela, Croatia[10]
Coastal Plan of the Split-Dalmatia County, Croatia [11]
National ICZM Strategy for Emilia-Romagna Region[5]

265 – 270 In addition, hard defences may also take different purposes, such as promenades, sunbathing platforms, roads, or parking places.[12]
* * *
[6] https://adriadapt.eu/adaptation-options/coastal-setback/

[7] Lincke, D., Wolff, C., Hinkel, J., Vafeidis, A., Blickensd Ãrfer, L., Povh Skugor, D. (2020). The effectiveness of setback zones for adapting to sea-level rise in Croatia. Regional Environmental Change, 20(46). doi:10.1007/s10113-020-01628-3

[8] Rochette, J., Du Puy-Montbrun, G., Wemaëre, M., Billé, R. (2010). Coastal setback zones in the Mediterranean: A study on Article 8-2 of the Mediterranean ICZM Protocol. IDDRI. Analyses 05/2010. https://www.iddri.org/sites/default/files/import/publications/an_1005_article-8-2-iczm-protocol.pdf

[9] https://iczmplatform.org//storage/documents/pEoju2FqfXjzPoYBLsKZiD3o6ONBXxJ44RTWFt7P.pdf

[10] https://adriadapt.eu/case-studies/coastal-zone-management-plan-for-the-city-of-kastela-coastal-plan/

[11] https://www.pomorskodobro.dalmacija.hr/DesktopModules/Bring2mind/DMX/API/Entries/Download?language=hr-HR&Command=Core_Download&EntryId=11315&PortalId=4

[12] https://adriadapt.eu/adaptation-options/seawalls-and-quays/

---

## Editor Comment (EC1)

Editorial Comment SP-2023-35 (KH-SLR Chapter 4)

Dear authors

The 3 reviewers have raised some valuable points, some of which were shared by multiple responses. Overall the work is being applauded for its relevance, rigour and presentation. In addition – and as expected – topics and references are being suggested to add to this overview. Since this paper is a chapter in an assessment report, the reviews are preferably treated as welcome extensions of the material on which the assessment is based.

In that respect, the following topics are considered to be valuable additions to the paper:

- **Classification of adaptation strategies** Some generic strategies to cope with sea level rise are (gradual) upgrading of defense infrastructure or coastal restoration (reviewer #2) and the identification of "setback zones" (elaborated extensively in the review by Daria Povh Skugor). Both have a strong rooting in present-day practices (even via legislation) or may become appropriate or critically limited in the future. Also there may be grey areas between some of the adaptation categories, such as "managed realignment for habitat creation" or the consideration of beach nourishment as an "advance strategy" (Reviewer #2). Reviewer #2 and Fraile Jurado also suggest to label early warning systems and forecasting tools as a separate class, and to elaborate in more depth on its developments and (increasing) value for instance for control of dynamic flood defense systems and evacuation. This may be combined with other information tools such as public information via cartography as suggested by Fraile Jurado. These additional categories may be added or integrated in existing categories, and elaborated in the main text
- **Distinction between current-day and future** Where possible, a distinction between what is happening today (practice) versus what might happen in the future (theory), for example for the approaches listed in Section 2.2.3 (Reviewer #2)
- **Discussion of trade-offs** Reviewer #2 makes various comments on side-effects, trade-offs and ancillary goals, such as the fact that sand for beach nourishment is scarce, the need to consider salinization of ground water even when it's not a safety issue, etc. Where possible I recommend to elaborate on the suggestions raised, it will make the overview a bit more insightful.
- **Documentation of existing laws and plans** Daria Povh Skugor notes that a discussion of existing laws and plans is missing, and Fraile Jurado also notes that there is no reference to national coastal protection laws. Daria's comment primarily implies a reference to the Barcelona convention that arranges the appointment of setback zones. I don't know whether this reviewer is right that this is the only regional legal text in the world that makes such an arrangement, and I'm not sure it's feasible to systematically survey the existence of such texts elsewhere. A cross-reference to Chapter 5 is appropriate (as the authors already replied to the reviewer), but a reference to this legal framework in Chapter 4 seems to be appropriate as it specifically refers to adaptation measures discussed here.
- **Conclusion section** Fraile Jurado suggests to include a specific conclusion section. However I would tend to keep the current structure where a summary per basin is given as a general conclusion. You could introduce this section in a way that the reader will consider it as a "regional policy oriented" summary and conclusion section.

Furthermore, the following specific comments of the reviewers could be addressed:

- A reference to the bibliographic database used for the review would be welcome, including any considerations used to categorize the literature into for instance the adaptation strategy classes. Also a statement is recommended on the fact that the literature included is not a reflection of the large body of literature that does exist, as indicated by Fraile Jurado.
- I agree with Reviewer #2 that a reference to mitigation needs to be either elaborated more prominently (for instance when it is framed as a limit to adaptation), or removed to avoid distraction from the focus on adaptation.
- A reference to Box 1 and a clear description of its goal would be welcome
- Inclusion of additional literature as suggested by the referees is highly recommended (for instance on the engineering solutions raised by Reviewer #2, the cost/benefit arguments to address limits of engineering solutions, a reference to AdriAdapt suggested by Daria Povh Skugor, etc)
- Quite a few suggestions for textual improvements or explaining specific statements can readily be followed-up

---

## Author Comment (AC2)

| SEA LEVEL RISE IN EUROPE: ADAPTATION MEASURES AND DECISION MAKING PRINCIPLES | Section | Comment | Status | Author's response | Last update |
|---|---|---|---|---|---|
| **Referee 1 - 15 feb 2024** | | | | | |
| **Q1** | General | Structural Enhancements: I find the absence of a dedicated conclusions section, or a "summary for policymakers," to be a notable omission. While the abstract performs a similar function, a robust set of general ideas articulated in a few paragraphs would greatly assist the sometimes cursory reading by those outside the scientific community, such as public officials, journalists, etc. This addition could significantly enhance the document's accessibility and utility by distilling its key insights into actionable guidance. | Done | We have written a conclusion to enhance the document accessibility | 4 April |
| **Q2** | 2.1 Adaptation measures | Development of Predictive Tools: The manuscript mentions the development of tools to forecast the scope and impacts of sea-level rise, yet this critical element seems to warrant further elaboration. The volume of scientific work on this topic, particularly regarding flooding, is remarkable. While it is not necessary to cite numerous studies, it is important to acknowledge that a significant portion of scientific efforts has focused on this issue for decades. This topic is likely one of the most published within the context of SLR. | Done | We added a statement on the focus of predictive tools regarding floodings. And we have added the role of predictive tools and cartografic techniques as far of trasversal responses to sea level rise. | 4 April |
| **Q3** | General | Cartographic Techniques: Following the previous comment, exploring cartographic techniques that can simply represent the implications of sea-level rise holds tremendous communicative potential for raising risk awareness. This potential tool for awareness is not mentioned in the report but could be one of the most powerful means of communication. | Done | We agree that maps are great tools for awareness raising, but tools for awarness raising do not fall within the scope of this paper, which focuses on decisions principles. In any case, we have added the role of predictive tools and cartografic techniques as far of trasversal responses to sea level rise. | 4 April |
| **Q4** | General | Compliance with Coastal Laws: There is a minor oversight, easily rectified by a simple sentence, regarding the compliance with coastal laws by states. This issue has often been overlooked in most scientific works on SLR, yet it is a critical aspect of addressing and mitigating the impacts of sea-level rise. | Done | We have added a statement regarding the compliance with coastal laws by states. | 4 April |
| **Q5** | 2.1 Adaptation measures | Formal Consideration of References: Lastly, on a minor formal note, the references to Spanish agencies should be reviewed. Typically, such references have been published by the Ministry of Environment or similar bodies, whereas the text currently attributes authorship to the title of the work itself. This should be revised in accordance with the editorial policies. | No action needed in the manuscript - we counterchecked with Lavinia and the handling editor (Bart) | The references were done according to the journal citation rules | 4 April |
| | | | | | |
| **Referee 2 - 18 Feb 2024** | | | | | |
| **Q6** | Section 2.2.3 Considering multiple criteria and interests | It is important to distinguish between what is happening today (practise) and aspirations for the future (theory). For example, in Section 2.2.3 it is unclear how much these approaches are used today versus this is a recommendation for the future. | Done | This comment is very well taken and we have improved the text to make this clear. | 4 April |
| **Q7** | 2.1 Adaptation measures | What are the strengths and limits of the literature review? One weakness is that adaptation measures are poorly reported in peer-reviewed literature so how representative is the sample? For example, could the review be enhanced by considering the grey literature? | Done | We further explained the methodology process, detailing all the steps undertaken and including an explanatory figure in this regard. | 4 April |
| **Q8** | 2.1 Adaptation measures | On line 117 you say: 'Although the literature examines in depth each type of response to sea level rise, accommodation measures are the most widely identified, followed by protection measures, advance measures, and finally retreat measures.' What metric is used to come to this judgement? Based on my experience, I think the dominant adaptation to date has been upgraded coastal defences, reflecting that Europe has a large legacy of coastal defences. Sea-level rise has been considered in defence upgrades around the southern North Sea for 20 to 30 years – that is a lot of activity and investment in the Netherlands, Germany, Belgium and the UK, but it is not written about at the scale of each scheme? The author's comments on this perspective is welcome if they have a counter argument. Is there public supplemental data on the literature review analysis? It would be useful to have the identified source and the analysis as a legacy to support future scholarship in this topic. Are the adaptation categories used unambiguous and are there any grey areas between them? | Done | 1. The statement on line 117 refers to the analysed documents (please check the explanatory table which has been added in the new version). 2 We have included upgrade defence under hard defence measures. 3There are grey areas between adaptation categories and we've stated in the chapter this as a limitation. In order to address this comment we have more explicitly stated across the text which measures could belong to other categories (early warning systems, adaptation of groundwater management, inegration of sea level rie in coastal adaptation strategies and plans, developping a risk culture within the population). 4. We have added a disclaimer stating the limitations of the research | 4 April |
| **Q9** | 2.1 Adaptation measures | I have already mentioned the large legacy of protection in Europe and especially around the North Sea. The paper tends to talk at times like we have a blank slate when there are thousands of kilometres of dikes and millions of people who depend on them to live as they do today. Negative aspects of dikes are raised, but as they already exist these are tempered somewhat. The main question is how they will be raised/upgraded in the future and where a more radical change might be necessary? Of course, new dikes may be built in some locations – I do not have a feel for how large this need might be and welcome the authors insights about the relatively importance of both. | Done | This is a very good point. We have added some text describing where coastal hard protection currently exists and also on the feasibility of upgrading coastal protection in the future under sea-level rise in the section about hard defences. | 4 April |
| **Q10** | Box 1: The MOSE system for protecting Venice and its lagoon | Box 1 – this is not referenced in the main text and the content is significant in size – I am unclear of the goal of this box? Also how useful is this box as Venice is a rather special case compared to Europe's coast? | Done | Goal of the box added and references to it in the text | 4 April |
| **Q11** | 2.3 Summary: key developments per basin | Section 3 Summary: key developments per basin – a key section –would be good to further develop specific Europe and European region recommendations. | Done | We have added recommendations for Europe in the concluding paragraph that will be submitted in the next version. | 4 April |
| **Q12** | General | The English could be tightened in general. | Done | English checked | 4 April |
| **Q13** | Introduction | Line 51 to 52 '. Traditional engineering solutions, here referred to as grey options, have dominated thinking and practice in coastal protection for several decades (Sancho, 2023).' – I would say they may have dominated for centuries – not to say there were not failures – but the defences were always rebuilt more strongly. For example see Kraus (1996) for many national portraits, or for a good national example -- the Netherlands (Van Koningsveld et al., 2008). | Done | Indeed, engineering protections have long dominated, as described in these papers you mentioned. However, here we are discussing a shorter time scale since we are considering the more recent risk of sea level rise due to climate change. The documents were read in any case (Kraus 1996: Van Koningsveld et al., 2008) | 4 April |

| SEA LEVEL RISE IN EUROPE: ADAPTATION MEASURES AND DECISION MAKING PRINCIPLES | Section | Comment | Status | Author's response | Last update |
|---|---|---|---|---|---|
| Q14 | 2.1 Adaptation measures | Line 99 Are 'early warning systems' just an accommodation response? They work for the entire coastline and are fundamental to mobile protection defences like the Thames Barrier and MOSE barrier in Venice. I would argue that they are an example of 'information measures' that are becoming progressively more powerful and useful and support all adaptation measures to varying degrees. | Done | We agree and we have more explicitly stated that early warning systems can support all type of responses. We have kept the original categorization, including that of 'technological option' rather than 'information and raise awareness option'. Altho it could belong to this later type of option, we have preferred to keep it under technological because this category encompasses more aims of early warning systems (forecasting, communication, etc.). | 4 April |
| Q15 | 2.1 Adaptation measures | Line 104 to 106 – under retreat where does managed realignment for habitat creation sit – a common action over the last 20 years. It appears later, but suggest it is worthy of definition here. | No action needed in the manuscript | Unclear comment | 4 April |
| Q16 | 2.1 Adaptation measures | Line 123 Table 1 – seems like the SROCC report is quite an influential source here – suggests this is a rather high level analysis rather than looking at practise in Europe. Line 123 Table 1 -- 14 'Rising and advancing coastal land' – I would say 'Raising and advancing coastal land'. Line 123 Table 1 -- 14 'Rising and advancing coastal land' – what insights do global analyses such as Sengupta et al (2023) provide on Europe? Line 123 Table 1 – 15 Beach and shoreface nourishment – is this really an advance strategy? If you introduce enough sand or gravel – yes? And maybe we advance in some areas because of an historical legacy of building in hazardous places. But sand and gravel are scarce and expensive so in general I think the goal is to hold the line. So I think this option is more nuanced than presented. Line 123 Table 1 – 16 Planned relocation – where is coastal restoration? | Done | As for the first point: regarding the frequent mention of the SROCC, it provided us with a solid scientific foundation regarding the measures mentioned to ensure their robustness. We used this foundation to then study other scientific articles and legislation at the EU and national levels in Europe and provide concrete examples of successes or projects in Europe. In addition, as suggested, we have reconsidered the category of the measure beach and shoreface nourishment, it has been moved from advance to protect. Finally, a citation of Sengupta et al (2023) has been added as requested about advancing coastal land in ports. | 4 April |
| Q17 | 2.1 Adaptation measures | Line 161 – what about avoiding salinisation of groundwater – that is of concern with sea-level rise especially when combined with overuse of these resources. | Done | This topic was addressed in the paper about impacts, we have added a statement on this (checked with the handling editor, Bart). | 4 April |
| Q18 | 2.1 Adaptation measures | Line 181 – restricting development can be considered a form of retreat – e.g., building setbacks along eroding cliffs? | Done | We have moved the measure to retreat. | 4 April |
| Q19 | 2.1 Adaptation measures | Line 220 to 223 – is this only accommodation? This information is useful for all coastal adaptation – it provides information on how high to build defences or how far to retreat. | Done | As suggested, we mentionned that instruments can indeed support all types of measures. | 4 April |
| Q20 | 2.1 Adaptation measures | Line 240 to 242 – people behind defences can be evacuated when the early warning system suggests an event that might cause a defence failure – these types of responses are not well documented but happen in practise. | Done | We agree and we have more explicitly stated that early warning systems can support all type of responses. We have added also an example of how early warning systems may be used in other types of response (in particualr refered to Thames and Mose). | 4 April |
| Q21 | 2.1 Adaptation measures | Line 258 ' On the other hand, perceived intractability of climate change hinders the desire to adopt low-carbon behaviours (Xiang et al., 2019).' – true but is it relevant here – this is climate mitigation and the section is about climate hazard risks. I would delete as a distraction or if important move to another more relevant section on synergies with mitigation. | Done | Xiang, et al. 2019 has been removed | 4 April |
| Q22 | 2.1 Adaptation measures | Line 263-264 – what 'technical limits' to protection – If you give an engineer enough resources, modern engineering will be able to provide a defence. The limits line much more in cost and cost-benefit, finance and social acceptance and also if sea-level rise is rapid (ice sheet collapse) – areas where there is much less research. This has been stated by Hinkel et al (2018). As an example to defend this view – the Thames Estuary Project planned defences of London against rises up to 5 m of sea-level rise (Tarrant and Sayers, 2012; Ranger et al., 2013). | Done | Your recomendations and suggested literature have been integrated in the text. | 4 April |
| Q23 | 2.1 Adaptation measures | Line 291 -- Restoration and management of coastal ecosystems are more widespread than listed with good examples in UK and Germany. Also how much are these strategies adaptation and how much are they coastal restoration which is complementary but not necessarily adaptation? | Done | We added an example in the UK | 4 April |
| Q24 | 2.1 Adaptation measures | Line 320 – considering coastal and shoreface nourishment as an example of Advance seems incorrect – if enough sediment is used it might be the case. However, as erosion is the overwhelming trigger of this strategy and advance is short-term at best and most of these projects aspire to stabilise the shoreline rather than advance. Later you state nourishment is not sustainable contradicting earlier remarks. The national scale nourishment of the Netherlands where erosion is outlawed is not mentioned. | Done | As suggested, we have reconsidered the category of the measure beach and shoreface nourishment, it has been moved from advance to protect. The last part regarding The Netherlands is not clear. | 4 April |
| Q25 | 2.1 Adaptation measures | Line 342 – references to support this statement? | Done | The reference of the initial sentence corresponds to the same source as the subsequent sentence (Pinto et al., 2020). | 4 April |
| Q26 | 2.1 Adaptation measures | Line 364 – what is the difference between managed relocation and managed realignment – seem to be rather similar terms to me. Managed relocation and retreat seems to be something that is going to happen while earlier sections of discussed active projects – this is an important distinction is not made explicit. | Done | Managed relocation, which assumes that migration takes place earlier due to its proactive initiation and supervision by governments, is similar to what we have here called 'planned relocation'. While managed realignment is a measure that usually results in the creation of a salt marsh by removing costal protection an allowing for an area previously protected from flooding to become flooded. We have clarified the confusion between managed relocation and managed realignment by adding managed realignment in a separate measure. | 4 April |
| Q27 | 2.1 Adaptation measures | Page 15 "divided into 4 arrows" – don't understand? | Done | Checked with author, right word added 'arrays'. | March 16 |

| SEA LEVEL RISE IN EUROPE: ADAPTATION MEASURES AND DECISION MAKING PRINCIPLES | Section | Comment | Status | Author's response | Last update |
|---|---|---|---|---|---|
| Q28 | Box 1: The MOSE system for protecting Venice and its lagoon | Page 17 "Venice constantly raised the building construction levels, to cope with SLR" – this should refer to relative SLR as subsidence is the main historic driver. | Done | Corrected: "Venice constantly raised the building construction levels, to cope with relarive SLR (due to eustacy+subsidence)" | March 4th |
| Q29 | 2.1.2 Limits and trade-offs of adaptations measures | Line 386-387 – as much of this protection is in place I am not sure this is conveying the present choices. | No action needed in the manuscript | Unclear comment | April 4 |
| Q30 | Box 3: Sea Level Rise and World Heritage Sites: the case of Wadden Sea | Line 467-469 – these SLR scenarios for the Wadden Sea seem rather precise – are the uncertainties being conveyed? – see line 557-559. | Done | Values represent the 5%-95% uncertainty range and are based on the reference indicated (Vermeersen et al., 2018). However, I would prefer not to add this information in the text as it is the most common way of reporting SLR (and readers can still go to the cited reference for further information) | March 1st |
| Q31 | Section 2.2.4 Implementation of low regret measures | Line 610-612 – what was the response to Xynthia in France – was there any retreat? | Done | Thnaks. This is indeed a good European example. I addded a few lines on this. | March 15 |
| Q32 | Section 2.2.6 Factoring SLR into decisions that need to be made today | Line 677-689 – the Thames Estuary 2100 project provides a real-world example of considering high-end SLR scenarios (e.g., Ranger et al., 2013) and might be used here. | Done | Indeed! We have addded this example. | March 15 |
| Q33 | General | References

Hinkel, J., Aerts, J.C.J.H., Brown, S. et al. 2018. The ability of societies to adapt to twenty-first-century sea-level rise. Nature Clim Change 8, 570–578, https://doi.org/10.1038/s41558-018-0176-z

Kraus, N.C. ed., 1996, June. History and heritage of coastal engineering. American Society of Civil Engineers, New York.

Ranger, N., Reeder, T. and Lowe, J., 2013. Addressing 'deep' uncertainty over long-term climate in major infrastructure projects: four innovations of the Thames Estuary 2100 Project. EURO Journal on Decision Processes, 1(3-4), pp.233-262.

Sengupta, D., Choi, Y.R., Tian, B., Brown, S., Meadows, M., Hackney, C.R., Banerjee, A., Li, Y., Chen, R. and Zhou, Y., 2023. Mapping 21st century global coastal land reclamation. Earth's Future, 11 (2), p.e2022EF002927.

Tarrant, O. and Sayers, P.B., 2012. Managing flood risk in the Thames Estuary–the development of a long-term robust and flexible strategy. In Flood risk: planning, design and management of flood defence infrastructure (pp. 303-326). ICE publishing.

Van Koningsveld, M., Mulder, J.P., Stive, M.J., Van Der Valk, L. and Van Der Weck, A.W., 2008. Living with sea-level rise and climate change: a case study of the Netherlands. Journal of Coastal Research, 24(2), pp.367-379 | Done | Some of the references were added | April 4 |